# EXPECTED ATTENTION:
# KV CACHE COMPRESSION BY ESTIMATING ATTENTION FROM FUTURE QUERIES DISTRIBUTION

## ABSTRACT

Memory consumption of the Key-Value (KV) cache represents a major bottleneck for efficient large language model (LLM) inference. While attention-score-based KV cache pruning shows promise, it faces critical practical limitations: attention scores from future tokens are unavailable during compression, and modern implementations do not materialize the full attention matrix, making past scores inaccessible. To overcome these challenges, we introduce *Expected Attention*, a training-free compression method that estimates Key-Value (KV) pairs importance by predicting how future queries will attend to them. Leveraging the distributional properties of LLM activations, we compute expected attention scores in closed form for each KV pair. These scores enable ranking and pruning of KV pairs with minimal impact on the residual stream, achieving high compression without performance degradation. Importantly, our method operates seamlessly across both prefilling and decoding phases, consistently outperforming state-of-the-art baselines in both scenarios. Finally, we release a comprehensive research library for KV cache compression, designed to enable researchers to implement and benchmark novel methods, in addition to building upon our own.

## 1 INTRODUCTION

Large language models (LLMs) (Achiam et al., 2023; Anthropic, 2025; MetaAI, 2024; Yang et al., 2025) have revolutionized text generation and reasoning, enabling advanced applications such as long multi-round dialogues, extensive multimodal intelligence (Yang et al., 2025; Weng et al., 2024), and agentic workflows that ingest massive amounts of data (OpenAI, 2024; PerplexityAI, 2025; Yamada et al., 2025). These applications often require processing extensive contextual information. For example, processing a large codebase or a short video can easily involve analyzing hundreds of thousands of tokens. A critical issue in deploying LLMs in such scenarios is the prohibitive memory consumption of the Key-Value (KV) cache (Fu, 2024; Shi et al., 2024; LI et al., 2025).

During autoregressive generation, the KV cache stores key and value vectors for every processed token, enabling efficient attention computation. However, its memory footprint grows linearly with sequence length, quickly becoming the primary bottleneck for long-context inference. A medium-sized 70B model (MetaAI, 2025) requires approximately 320 GB of GPU memory for a one-million-token KV cache, far exceeding most GPU capacities. This challenge intensifies with emerging applications where advanced reasoning models generate thousands of intermediate tokens (DeepSeek-AI, 2024b; Yang et al., 2025) and agentic systems load massive datasets (OpenAI, 2025; PerplexityAI, 2025). While current LLMs promise extended context lengths up to a million tokens (GeminiTeam, 2025; MetaAI, 2024), hardware constraints saturate GPU memory well before reaching theoretical limits.

State Space Models offer a solution by reducing memory costs (Gu et al., 2022; Gu & Dao, 2024), yet their inferior performance compared to transformers, especially on long context tasks, limits adoption (Jelassi et al., 2024; Merrill et al., 2024). Other architectural changes limited to the attention mechanism, such as multi-head latent attention (DeepSeek-AI, 2024a) or sliding window attention (Jiang et al., 2023; GemmaTeam, 2025), reduce KV cache size but do not remove the attention bottleneck and are orthogonal to KV cache compression methods. Additionally, such methods need

to be implemented at training time, limiting their application to pre-trained modern LLMs. This creates demand for training-free KV cache compression methods that preserve transformer architectures while mitigating memory growth.

KV cache compression exploits semantic redundancy in natural language: not all tokens equally influence future predictions, and many provide negligible information once their contextual role is fulfilled. This property allows to compress the KV cache by removing some of the key and values stored in it. However, determining which tokens can be safely removed is far from trivial, as any Key-Value (KV) pair's importance depends on how *future queries* will attend to it. Existing approaches use heuristics like discarding oldest tokens (Ge et al., 2024; Xiao et al., 2023) or leverage attention scores from past queries (Zhang et al., 2024; Li et al., 2025; Oren et al., 2024), but these strategies are limited for real-world scenarios, and often require accessing attention scores which are not materialized in modern transformer implementations (Dao et al., 2022).

Instead of relying on heuristics or local attention metrics, we argue that a KV pair's significance is best measured by its global effect on the transformer's output. We quantify this effect by isolating each KV pair's contribution within the residual stream, capturing its influence on the model output. This raises the challenge of estimating *how future queries will attend to each token in the context*, which requires accessing attention scores from the past and from future tokens, that are not available at the time of compression. To address this, we introduce *Expected Attention*, which estimates future attention allocation leveraging the distribution of future queries. Expected Attention estimates the importance that each token in the context has for queries that have not been generated and accordingly prunes the KV cache up to 60% while preserving performance quality, requiring no architectural modifications or additional training. We release our code as a comprehensive library benchmarking over 20 state-of-the-art compression methods.

To summarize, our contributions are the following:

- We analyse the distributional properties of LLM activations through the lenses of KV cache compression and introduce the concept of *Expected Attention* to estimate the importance that current tokens will have in the future.
- We introduce a KV cache compression method that leverages Expected Attention and evicts irrelevant KV pairs for efficient inference.
- We release all our code as a library, designed for researchers, that allows to easily implement, test and benchmark KV cache compression methods.

## 2 EXPECTED ATTENTION

### 2.1 KEY-VALUE CACHE IN AUTOREGRESSIVE TRANSFORMERS

We consider decoder-only language models based on the transformer architecture (Vaswani et al., 2017), representing the vast majority of modern LLMs. When an input sequence of tokens $\mathbf{x} = [x_1, x_2, \ldots, x_t]$ is fed to the model, each token $x_i$ is transformed into a hidden state representation $h_i \in \mathbb{R}^h$ and processed by a stack of transformer layers, including feed forward networks and multi-head attention blocks. For brevity and clarity, we focus our analysis on a single layer and attention

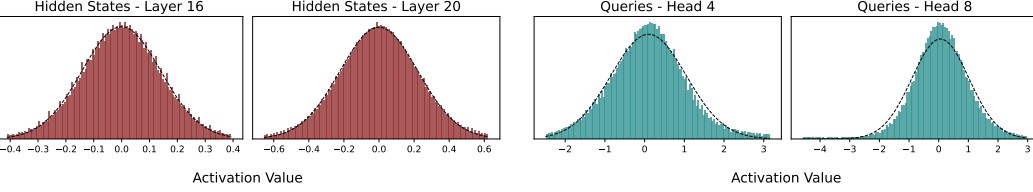

Figure 1: Hidden states from layer 16 and 20 and corresponding queries for layer 20 in Llama3.1-8B. Hidden states in modern LLMs are mostly normally distributed. As a consequence, query activations also follow a Normal. The best Gaussian fit is overlayed. We show more examples and discuss this property in Appendix B.

head, noting that the following analysis naturally extends to multi-head attention, grouped query attention (GQA, Ainslie et al. 2023) and all their variants.

Let $h_i \in \mathbb{R}^h$ denote the hidden state at position $i$ in the sequence. In the attention block, the corresponding Query, Key and Value projections are computed as:

$$q_i = R_i W_Q h_i, \quad k_i = R_i W_K h_i, \quad v_i = W_V h_i \tag{1}$$

where $d$ is the attention head dimension, $R_i \in \mathbb{R}^{d \times d}$ is the Rotary Position Embedding (RoPE, Su et al. 2023) matrix at position $i$, and $W_Q, W_K, W_V \in \mathbb{R}^{h \times d}$ are respectively the learnable projection matrices for query, key, and value in $\mathbb{R}^d$. During autoregressive inference, keys and values vectors are stored in the KV cache to avoid recomputing them in future generation steps. The resulting KV cache is a collection of Key-Value pairs $(k_i, v_i)$ from all inference steps in the sequence, leading to significant computational savings but increasing memory requirements, growing linearly with sequence length.

At generation step $t$, the attention mechanism computes the attention score between the current query $q_t$ and each previously cached key $k_i$ for $i \leq t$:

$$a_{ti} = \frac{\exp\left(\frac{q_t^T k_i}{\sqrt{d}}\right)}{\sum_{j=1}^t \exp\left(\frac{q_t^T k_j}{\sqrt{d}}\right)} = \frac{z_{ti}}{\sum_{j=1}^t z_{tj}} \tag{2}$$

where $a_{ti}$ is the normalized attention score between query at position $t$ and key at position $i$, and $z_{ti} = \exp\left(\frac{q_t^T k_i}{\sqrt{d}}\right)$ represents the unnormalized attention score.

The attention score is used to weight and sum over all values previously stored in the KV cache. The resulting output is then added to the hidden state $h_t$:

$$h_t^{\text{out}} = h_t + \sum_{i=1}^t a_{ti} W_o v_i = h_t + \sum_{i=1}^t \Delta h_{ti} \tag{3}$$

where $h_t \in \mathbb{R}^h$ and $h_t^{\text{out}} \in \mathbb{R}^h$ represent the hidden state before and after the attention update respectively, and $W_o \in \mathbb{R}^{d \times h}$ is the learnable output projection matrix. The hidden states embedding $h_t$ represents the "residual stream," (Elhage et al., 2021) updated via vector additions by each transformer block. The value $\Delta h_{ti} = a_{ti} W_o v_i$ isolates the specific residual addition of the $i$-th KV pair at step $t$. This decomposition reveals that each cached KV pair $(k_i, v_i)$ contributes a residual update $\Delta h_{ti}$ to the final output, and provides a natural measure of the importance of each KV pair:

$$\|\Delta h_{ti}\| = a_{ti} \|W_o v_i\| \tag{4}$$

where $\| \cdot \|$ denotes the L2 norm. This metric captures both the attention weight $a_{ti}$ (how much the query attends to the $i$-th key) and the transformed value magnitude $\|W_o v_i\|$ (the impact of the $i$-th value on the output). Equation 4 provides the optimal measure for estimating the importance of each KV pair in the model output. If we could compute this score for all cached KV pairs, we could selectively prune the cache by removing pairs with the lowest impact on the residual stream, thereby minimizing performance degradation. However, computing Equation 4 presents significant practical challenges. While $\|W_o v_i\|$ is readily available at inference time, the attention weight $a_{ti}$ depends on future queries that have not yet been generated. Specifically, we cannot know the attention scores from future tokens $t+1, t+2, \ldots$ before computing them, making it impossible to predict which KV pairs will be important for upcoming generation steps. Furthermore, modern transformer implementations utilize Flash Attention (Dao et al., 2022; Dao, 2024), which computes attention scores on-the-fly without materializing the complete attention matrix, preventing access to even past attention scores. To address these fundamental limitations, we leverage the properties of activations in modern LLMs, and introduce *Expected Attention*.

## 2.2 EXPECTED ATTENTION: ESTIMATING ATTENTION FROM FUTURE QUERIES

**Distributional properties of LLM activations** To approximate the unnormalized attention score $z_{ij}$, we leverage the findings of Liu et al. (2025), showing that hidden states in modern LLMs loosely follow a Gaussian distribution $h \sim \mathcal{N}(\mu, \Sigma)$. While we show an example of this property

in Figure 1, we also extensively validate it across multiple model architectures in Appendix B. Given this distributional assumption, queries also inherit unimodal properties through the linear transformation in Equation 1 $q_t = R_t W_Q h_t$, and can be approximated with a Gaussian (Liu et al., 2025):

$$q_t \sim \mathcal{N}(\mu_{q_t}, \Sigma_{q_t}), \quad \text{where } \mu_{q_t} = R_t W_Q \mu, \quad \Sigma_{q_t} = R_t W_Q \Sigma W_Q^T R_t^T \tag{5}$$

where $\mu \in \mathbb{R}^d$ and $\Sigma \in \mathbb{R}^{d \times d}$ are the mean and covariance of the hidden state distribution, and $R_t \in \mathbb{R}^{d \times d}$ is the RoPE matrix at position $t$.

To create a single, tractable representation of attention over a future interval, we approximate the positional embeddings by averaging the RoPE matrix over the next $T$ positions. This gives us a position-averaged query distribution:

$$\bar{q} \sim \mathcal{N}(\bar{\mu}_q, \bar{\Sigma}_q), \quad \text{where } \bar{\mu}_q = \bar{R} W_Q \mu, \quad \bar{\Sigma}_q = \bar{R} W_Q \Sigma W_Q^T \bar{R}^T \tag{6}$$

where $\bar{R} = \frac{1}{T} \sum_{j=1}^{T} R_{t+j}$ represents the averaged RoPE matrix over $T$ future positions.

```python
def compress(queries, keys, values, compression_ratio):
    # Compute query statistics
    mean_query, cov_query = compute_statistics(queries)
    # Compute unnormalized attention scores (z_i)
    scores = matmul(mean_query, keys.T) / math.sqrt(d)
    scores += einsum("i,ij,j->", keys, cov_query, keys) / (2 * d)
    # Normalize scores and weight by value norms
    scores = softmax(scores, dim=-1) * values.norm(dim=-1)
    # Keep KV pairs with highest scores
    n_kept = int(keys.size(0) * (1 - compression_ratio))
    indices = scores.topk(n_kept, dim=-1).indices
    return keys[indices], values[indices]
```

Listing 1: Pytorch-like pseudo code for KV Cache compression with Expected Attention.

**Expected Attention Score**  With this query distribution, we can now analytically compute the expected unnormalized attention score in Equation 2. For a query $\bar{q} \sim \mathcal{N}(\bar{\mu}_q, \bar{\Sigma}_q)$ in our interval $T$ and a fixed key $k_i$, the expected unnormalized score for that key is:

$$\hat{z}_i = \mathbb{E}_{\bar{q} \sim \mathcal{N}(\bar{\mu}_q, \bar{\Sigma}_q)} \left[ \exp\left( \frac{\bar{q}^T k_i}{\sqrt{d}} \right) \right] = \exp\left( \frac{\bar{\mu}_q^T k_i}{\sqrt{d}} + \frac{k_i^T \bar{\Sigma}_q k_i}{2d} \right) \tag{7}$$

where the second equality follows from the moment-generating function of a Gaussian distribution. We then define the expected attention score by applying the softmax on our unnormalized expectation:

$$\hat{a}_i = \frac{\hat{z}_i}{\sum_{j=1}^{t} \hat{z}_j} \tag{8}$$

With this approximation, we can now estimate the importance of each cached KV pair. We define the expected contribution magnitude by substituting our expected attention weight into the contribution score formula from Equation 4:

$$\|\widehat{\Delta h_i}\| = (\hat{a}_i + \epsilon)\|W_o v_i\| \tag{9}$$

where $\hat{a}_i$ is the expected attention weight from Equation 8, $\|W_o v_i\| \in \mathbb{R}$ is the magnitude of the transformed value vector, and $\epsilon$ is a small hyperparameter. This metric provides a tractable approximation to the true contribution score without requiring future queries.

**Compression with Expected Attention**  Equation 9 captures the contribution of each KV pair to the transformer output. The Expected Attention compression algorithm scores all cached KV pairs according to Equation 9 and evicts the $r\%$ pairs with the lowest expected contributions, where $r \in [0, 1]$ is the compression ratio. Intuitively, this is equivalent to removing those KV pairs that have the smallest impact on the residual stream and therefore on the model output. We provide pseudo-code for our compression algorithm in Listing 1.

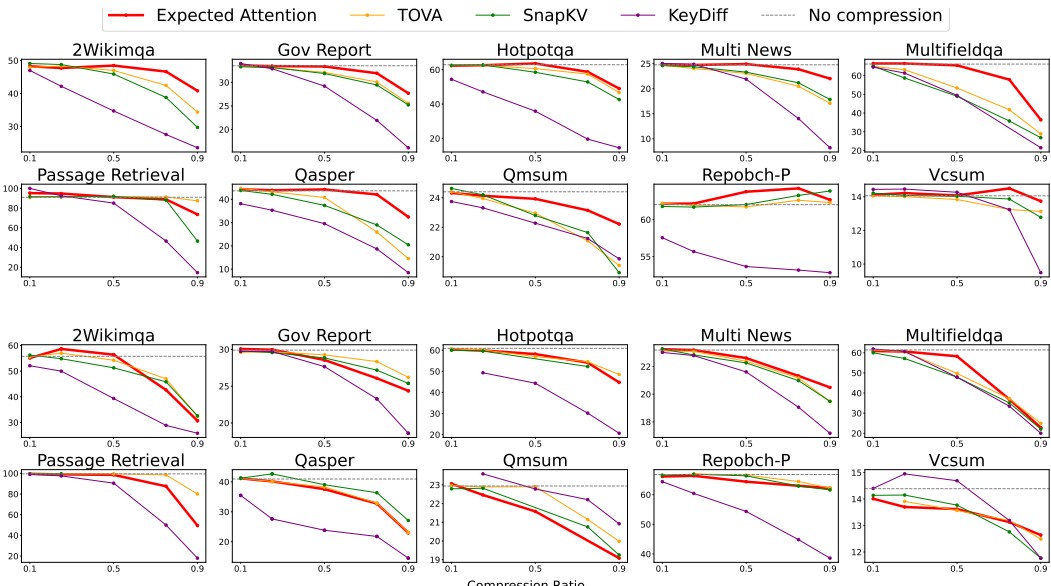

Figure 2: Scores on LongBench (Bai et al., 2024) for Qwen3-8B (top) and Gemma3-12B (bottom). The x-axis represents the compression ratio, the y-axis the score for each specific dataset. The horizontal line represents the baseline performance without cache compression. Expected Attention achieves optimal trade-off between compression ratio and scores across most datasets (Additional and averaged results in Appendix E).

**Head-Adaptive Compression**  Previous work has shown that different attention heads serve different roles in the model. We adopt adaptive per-layer compression (Feng et al., 2024) to account for this heterogeneity, allowing more important heads to retain more KV pairs.

## 3 EXPERIMENTS

### 3.1 EXPERIMENTAL SETUP

**Prefilling vs Decoding Generation**  LLM inference comprises two phases with distinct computational characteristics. The *prefilling phase* processes the entire input prompt in parallel, computing key-value projections for the KV cache, a compute-bound operation requiring substantial floating-point operations. The *decoding phase* sequentially generates tokens using the KV cache and previous logits, appending new key-value pairs iteratively (Deepak & Amr, 2024; Gordić, 2025). This dichotomy has motivated disaggregated architectures that implement prefill and decoding on different hardware (Deepak Patil, 2024; StepFun et al., 2025), at the cost of transferring the cache, further incentivising compression. An effective compression method must perform well in both prefilling and decoding (Deepak & Amr, 2024; Gordić, 2025). Nevertheless, a number of recent methods often target a single phase: SnapKV (Li et al., 2025) for prefilling via query attention scores, StreamingLLM (Xiao et al., 2023) and KNorm (Devoto et al., 2024) for streaming decoding. Expected Attention is designed considering these two aspects of LLM inference and addresses both scenarios efficiently. We present results for prefilling and decoding in Section 4.1 and Section 4.2 respectively.

**Models and Datasets**  For prefilling (one-shot compression before generation), we test three model families supporting long contexts: Llama3.1-8B (128k) (MetaAI, 2025), Qwen3-8B (32k) (Yang et al., 2025), and Gemma3-12B (128k) (GemmaTeam, 2025), all instruction-tuned. For decoding (compression during generation), we analyse reasoning models that generate extensive intermediate reasoning tokens and therefore large KV caches: Qwen-1.5B-R1, Qwen-7B-R1 (DeepSeek-AI, 2025), and OpenMath-Nemotron-14B (Moshkov et al., 2025).

Table 1: Expected Attention outperforms most baselines on Ruler (Hsieh et al., 2024) with 4K and 16K context length. We show average score with increasing compression ratios across baselines. Best results for each compression ratio are displayed in **bold**. The 0% column indicates the baseline without compression.

| Model | Method | Ruler 4k | | | | | | Ruler 16k | | | | | |
|---|---|---|---|---|---|---|---|---|---|---|---|---|---|
| | | 0% | 10% | 25% | 50% | 75% | 90% | 0% | 10% | 25% | 50% | 75% | 90% |
| Qwen3-8B | EA (ours) | **95.3** | **95.3** | **95.0** | **94.7** | **88.3** | **65.4** | **92.9** | **93.1** | **93.2** | **92.7** | **85.6** | **62.7** |
| | TOVA[51] | **95.3** | 89.0 | 82.5 | 77.6 | 62.4 | 24.7 | **92.9** | 88.3 | 81.7 | 76.2 | 68.7 | 52.4 |
| | SnapKV[38] | **95.3** | 92.6 | 84.0 | 55.7 | 33.1 | 19.2 | **92.9** | 90.1 | 81.5 | 62.8 | 41.7 | 26.8 |
| | KeyDiff[52] | **95.3** | 93.8 | 89.4 | 78.6 | 64.4 | 37.9 | **92.9** | 88.9 | 82.9 | 74.5 | 66.9 | 53.1 |
| Gemma3-12B | EA (ours) | **95.2** | **95.2** | **94.9** | **92.7** | **78.2** | **53.6** | **86.0** | **82.8** | **81.7** | **76.6** | **60.5** | **41.8** |
| | TOVA[51] | **95.2** | 89.7 | 81.1 | 76.5 | 58.1 | 25.3 | **86.0** | 79.7 | 72.6 | 62.5 | 46.8 | 32.7 |
| | SnapKV[38] | **95.2** | 82.9 | 72.0 | 54.8 | 40.3 | 30.1 | **86.0** | 74.1 | 62.8 | 46.4 | 37.3 | 31.4 |
| | KeyDiff[52] | **95.2** | 94.3 | 90.6 | 79.8 | 62.0 | 34.3 | **86.0** | 81.8 | 78.6 | 72.6 | 58.6 | 37.2 |
| Llama3.1-8B | EA (ours) | **95.3** | **95.7** | 95.3 | 92.2 | **75.9** | 30.6 | **93.4** | **93.4** | 92.8 | 86.0 | 66.4 | 25.5 |
| | TOVA[51] | **95.3** | 93.2 | 87.3 | 76.2 | 63.3 | 37.5 | **93.4** | 90.9 | 86.1 | 77.9 | 68.4 | 59.2 |
| | Duo [65] | **95.3** | **95.7** | **95.7** | **95.3** | 73.2 | 24.5 | **93.4** | 93.3 | **93.0** | **90.1** | 59.1 | 12.3 |
| | SnapKV[38] | **95.3** | 95.5 | 88.8 | 81.8 | 63.2 | 43.4 | **93.4** | 89.4 | 82.0 | 68.0 | 43.1 | 25.6 |
| | KeyDiff[52] | **95.3** | 94.7 | 91.6 | 85.5 | 72.9 | **61.1** | **93.4** | 92.1 | 88.4 | 82.6 | **74.9** | **66.5** |

Our benchmarks include LongBench (Bai et al., 2024), Ruler (Hsieh et al., 2024), and Needle in a Haystack (Kamradt, 2023; Liu et al., 2024) for prefilling, and Aime25 (Balunović et al., 2025) and MATH-500 (Lightman et al., 2023) for decoding.

**Baselines** Following an initial benchmarking study on Ruler (see Appendix E), we selected and compare our method against the best-performing baselines for each use case. For prefilling, we evaluate attention-based approaches like SnapKV (Li et al., 2025) and TOVA (Oren et al., 2024), embedding-based KeyDiff (Park et al., 2025), and the trainable DuoAttention (Xiao et al., 2024) when the checkpoint is available. SnapKV (Li et al., 2025) and TOVA (Oren et al., 2024) rank KV pairs using attention scores from user queries. KeyDiff (Park et al., 2025) employs distance metrics between key embeddings for selection, making it also suitable for decoding generation. DuoAttention (Xiao et al., 2024) takes a trainable approach, learning compression masks for each attention head. For decoding, we focus on methods designed to be compatible with streaming generation: KNorm (Devoto et al., 2024), StreamingLLM (Xiao et al., 2023), and KeyDiff (Park et al., 2025). KNorm (Devoto et al., 2024) uses a simple approach by preserving keys with the lowest $L_2$ norm. StreamingLLM (Xiao et al., 2023) maintains initial sink tokens throughout generation.

**Implementation details** We implement Expected Attention in Pytorch (Paszke et al., 2019). For all benchmarks, we test the models on 8 H100 GPUs, with batch size 1. We make all the code to reproduce our method and the baselines available online. In all experiments we use $\epsilon = 0.01$, except for needle in a haystack where use $\epsilon = 0$, and we average the RoPE embeddings over the next $T = 512$ positions. For prefilling, we do not assume any question about the context. This simulates a real world use case and avoids favouring methods like SnapKV that rely on this assumption. For decoding, we keep a small buffer of hidden states of 256 tokens to compute statistics, and perform compression every 512 generation steps. In Equation 9 we only use $V$ instead of $W_oV$, as using $W_o$ led to a minor increase in results at a significantly higher memory cost.

## 4 EXPERIMENTAL RESULTS

### 4.1 PREFILLING

**LongBench** We evaluate on LongBench (Bai et al., 2024), which tests long-context capabilities across diverse tasks. The benchmark comprises six categories: single and multi-document QA, summarization, few-shot learning, synthetic tasks, and code completion. As shown in Figure 2 for Llama3.1-8B and Qwen3-8B (see Appendix E for Gemma3-12B), Expected Attention consistently achieves optimal compression-performance trade-offs, maintaining higher scores across all compression ratios. This demonstrates effective retention of critical KV pairs even under significant compression across varied reasoning and generation tasks.

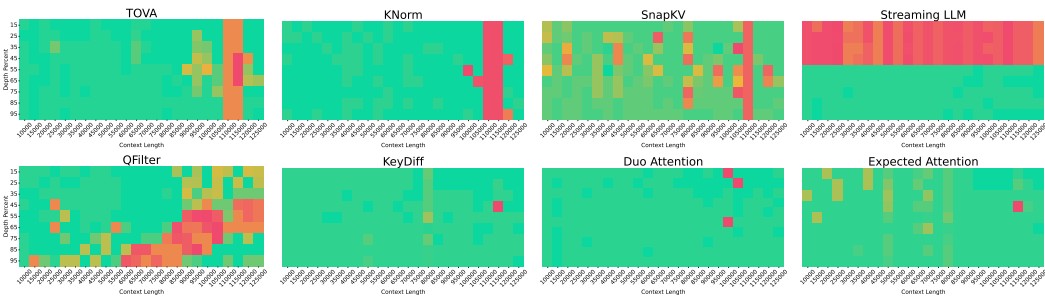

Figure 3: Needle in the Haystack test for different methods with Llama3.1-8B and 50% compression ratio.

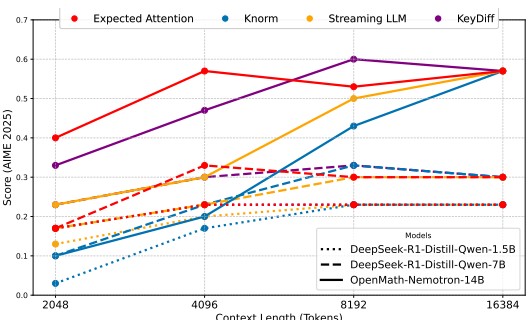

Figure 4: Decoding results on Aime25 dataset, different markers represent different models sizes. The x-axis is the maximum size that the KV cache is allowed to grow to.

Table 2: Decoding scores on MATH-500. Columns indicate the final size of the KV cache with respect to the original full version. Best scores in **bold**.

| Model | Method | Compression | | | |
|-------|--------|:---:|:---:|:---:|:---:|
| | | 0× | 2× | 4× | 12× |
| Qwen-R1-1.5B | EA (ours) | **0.47** | **0.47** | **0.43** | **0.33** |
| | KeyDiff[52] | **0.47** | 0.42 | 0.40 | 0.30 |
| | KNorm[15] | **0.47** | 0.41 | 0.28 | 0.11 |
| | Streaming[64] | **0.47** | 0.45 | 0.41 | 0.31 |
| Qwen-R1-7B | EA (ours) | **0.57** | **0.55** | **0.53** | **0.49** |
| | KeyDiff[52] | **0.57** | 0.54 | 0.48 | 0.35 |
| | KNorm[15] | **0.57** | 0.47 | 0.32 | 0.12 |
| | Streaming[64] | **0.57** | 0.54 | 0.51 | 0.41 |
| Nemotron-14B | EA (ours) | **0.57** | 0.55 | **0.54** | **0.47** |
| | KeyDiff[52] | **0.57** | 0.56 | 0.51 | 0.44 |
| | KNorm[15] | **0.57** | 0.50 | 0.36 | 0.14 |
| | Streaming[64] | **0.57** | **0.57** | **0.54** | 0.42 |

**Ruler** Ruler (Hsieh et al., 2024) measures retrieval, multi-hop tracing, and aggregation abilities within long contexts through four subsets: NIAH (Needle-in-a-Haystack) for single-fact retrieval, VT (Variable Tracking) for multi-hop reasoning, CWE (Common Words Extraction) for frequency-based aggregation, and FWE (Frequent Words Extraction) for statistical pattern recognition. Table 1 shows results at various compression ratios for 4k and 16k windows. EA maintains strong performance across all subsets, particularly at higher compression ratios. While KeyDiff performs well on Llama3.1-8B, it struggles on Gemma3-12B and Qwen3-8B, potentially due to QK normalization (GemmaTeam, 2025; Yang et al., 2025). We note that the competitive performance of KeyDiff is often isolated to the extreme 75-90% compression ratio, a regime that is not the intended operating point for practical KV cache compression, whose main goal is to keep the downstream performance as close as possible to the uncompressed baseline. Our Expected Attention-based policy effectively preserves information necessary for precise retrieval and complex reasoning tasks.

**Needle in a Haystack** The NIAH test (Kamradt, 2023) embeds specific information (the "needle") within lengthy distracting text (the "haystack") to evaluate retrieval capabilities across varying context positions and lengths. The test systematically varies both the needle's position within the context (needle depth) and the total context length to assess consistent retrieval performance. Figure 3 visualizes retrieval success across needle positions and context lengths up to 125k tokens. Expected Attention demonstrates robust performance comparable to DuoAttention and significantly more stable than other baselines in long contexts, confirming retention of critical information under compression regardless of needle placement or context size.

## 4.2 DECODING

We evaluate Expected Attention on reasoning models, Qwen-1.5B-R1, Qwen-7B-R1, and OpenMath-Nemotron-14B. Reasoning models are particularly suitable for our evaluation as they generate extensive chain-of-thought outputs, placing significant demands on KV cache mem-

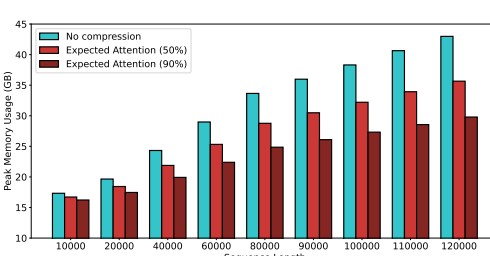 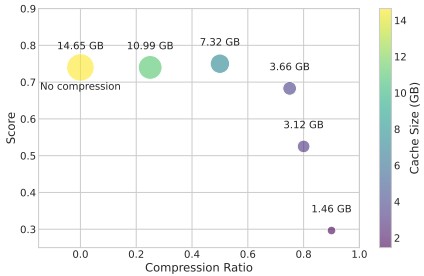

(a) Peak memory usage vs sequence length up to 120k for Llama3.1-8B, with 50% and 90% compression ratio. As the context length grows the memory savings become more evident, achieving up to 15GB less memory for large contexts.

(b) Needle in a Haystack score with different compression ratios with Qwen3-8B. Expected Attention has no accuracy loss with a compression ratio of 50%. Marker size indicates actual KV cache size in GB.

Figure 5: Memory footprint of Expected Attention with different compression ratios.

ory (Łańcucki et al., 2025). We use the Aime25 (Yamada et al., 2025) and MATH-500 (Lightman et al., 2023) datasets. Aime25 consists of competition-level mathematical problems requiring multi-step reasoning and precise calculation, while MATH-500 encompasses diverse mathematical domains including algebra, geometry, and number theory with varying difficulty levels. During decoding, we allow the KV cache to expand to a predetermined size before initiating token eviction. We use $n\times$ to show that the final cache size is $n$ times smaller than would be without compression.

Results for Aime25 and MATH-500 are presented in Section 4.1 and Table 2, respectively. EA consistently outperforms or matches baseline methods across all models, with particularly strong performance at higher compression ratios ($4\times$ and $16\times$). Most methods demonstrate minimal performance degradation at $2\times$ compression, indicating that a large portion of tokens in reasoning traces contains redundant information that can be pruned without affecting mathematical reasoning performance. Expected Attention shows the best performance especially in high-compression scenarios ($12\times$ compression).

## 4.3 MEMORY SAVINGS AND EFFICIENCY

We evaluate the memory efficiency of our method using Llama3.1-8B and Qwen3-8B for both prefilling and decoding phases. All experiments are conducted on a single H100 GPU with bfloat16 precision for both model weights and KV cache. We focus on peak memory usage as the primary efficiency metric, as KV cache memory consumption is often the primary bottleneck for long-context inference.

Figure 5a demonstrates peak memory usage as sequence length increases up to 120k tokens, comparing Expected Attention at 50% and 90% compression ratios against the uncompressed baseline with vanilla attention. The results show that memory savings become increasingly substantial as context length grows.

Figure 5b illustrates the relationship between compression ratio (x-axis) and NIAH benchmark performance for Qwen3-8B, with marker size representing the corresponding KV cache size. While higher compression ratios naturally reduce KV cache size, they typically incur performance penalties. Remarkably, Expected Attention at 50% compression maintains performance parity with the uncompressed baseline while achieving a $2\times$ reduction in KV cache size, demonstrating an optimal balance between memory efficiency and task performance.

## 4.4 LATENCY

**FLOPs Analysis** We provide a FLOPs analysis performed following Hoffmann et al. (2022), computing the total FLOPs required for the forward pass of Llama3.1-8B and then the additional FLOPs resulting from the Expected Attention overhead. Results show that the computational overhead

Table 3: Latency analysis with Llama3.1-8B for Prefilling, Generation and Total, when performing decoding on a 128K context with 50% compression. All results in seconds.

| Phase | No Compression | EA (50% compression) | Variation (abs) | Variation (%) |
|---|---|---|---|---|
| Prefilling | $15.25 \pm 0.02$ | $15.52 \pm 0.02$ | +0.27 | +1.74 % |
| Generation | $4.33 \pm 0.00$ | $3.22 \pm 0.04$ | -1.11 | -25.58 % |
| **Total** | $19.58 \pm 0.03$ | $18.74 \pm 0.03$ | -0.84 | -4.30 % |

accounts for just **0.5%** of the model's total FLOPs. This confirms that the theoretical increase in computational cost is negligible, validating the efficiency of our method. The complete methodology and derivation of these FLOPs are detailed in Appendix F.

**Empirical Latency Measurements**   To complement the theoretical analysis, we perform latency measurements using our PyTorch implementation. As summarized in Table 3, we achieve a **25%** reduction in generation latency due to the smaller cache footprint, which outweighs the $\sim 2\%$ prefill overhead. This results in a **4.3%** total latency reduction. Note that these measurements serve as an upper bound, as optimized kernels were not implemented and would further reduce the overhead.

## 5   ABLATION STUDIES

**Sensitivity to Future Window $T$**   We investigate the sensitivity of our method to the choice of the future window size $T$ used for the RoPE matrix approximation. As shown in Table 4a, the minimal performance drop observed across different models when reducing $T$ from 1024 to 512 or even 256 is justifying the practical choice of $T = 512$ that we used in our experiments.

**Adaptive Compression**   We conduct an ablation study against a uniform compression baseline (applying the same ratio to all heads) to assess its importance. The results in Table 4 show a significant performance drop for the uniform baseline, confirming that the adaptive approach is essential for retaining model accuracy.

**Covariance Term**   We investigate the contribution of the covariance term. While its removal causes a noticeable performance drop ($92.2 \rightarrow 90.6$) for Llama3.1-8B, the effect is minimal for Qwen3-8B and Gemma3-12B. We conjecture this reduced dependency is due to their QK normalization. This finding is particularly encouraging as it suggests that for models employing QK normalization, we could safely omit the covariance term in future implementations, thereby making the method even simpler.

Table 4: Ablation Study Results on Window Size, Adaptive Compression and Covariance

(a) Window Size $T$

| Model | T=1024 | T=512 | T=256 | T=128 |
|---|---|---|---|---|
| **Llama3** | 92.1 | 92.2 | 91.9 | 91.8 |
| **Qwen3** | 94.8 | 94.7 | 94.7 | 94.8 |
| **Gemma3** | 92.7 | 92.7 | 92.7 | 92.7 |

(b) Adaptive Compression and Covariance

| Model | EA | w/o Adaptive | w/o Covariance |
|---|---|---|---|
| **Llama3** | 92.2 | 86.5 | 90.6 |
| **Qwen3** | 94.7 | 86.6 | 94.7 |
| **Gemma3** | 92.7 | 88.2 | 92.6 |

## 6   RELATED WORKS

**Trainable KV-Cache Compression**   One approach to reducing memory requirements involves modifying the model architecture or training procedure to inherently produce smaller caches. Ainslie et al. (2023); Shazeer (2019) reduce cache size by decreasing the number of key-value heads, effectively sharing key-value representations across queries. DeepSeek-V2 (DeepSeek-AI, 2024b) introduced Multi-Head Latent Attention, which projects keys and values into a lower-dimensional latent space during training, directly reducing the memory footprint of cached representations. Alternative trainable approaches focus on learning compression policies (Łańcucki et al., 2025; Nawrot et al.,

2024) or masks (Xiao et al., 2024) from pre-trained checkpoints. Finally, State Space Models (Gu et al., 2022; Gu & Dao, 2024) replace the quadratic attention mechanism with linear-complexity alternatives, while hybrid approaches combine transformer layers with RNN-based components (Ren et al., 2025; Glorioso et al., 2024). Although these trainable methods typically achieve superior performance, they require substantial computational resources for pre-training or continued pre-training, making them less practical for deployment with existing large-scale models.

**Training-Free KV cache compression** Given the computational costs associated with trainable methods, significant research effort has focused on developing post-training compression techniques that can be applied to existing models without modification. Early approaches (Li et al., 2025; Oren et al., 2024) directly utilize attention scores to rank KV pairs by importance. However, these methods require access to the full attention matrix, making them incompatible with Flash Attention (Dao et al., 2022) and thus impractical for modern deployment scenarios. To address this limitation, several works have developed heuristic-based importance measures that can be computed without materializing attention matrices, such as keys norm (KNorm Devoto et al. (2024)), token positions (StreamingLLM Xiao et al. (2023), H2O Zhang et al. (2024)) or SVD projection (Q-Filters Godey et al. (2025)). Recognizing that different attention heads exhibit varying sensitivity to compression, recent methods such as AdaKV (Feng et al., 2024) and PyramidKV (Cai et al., 2025a) adopt head-specific compression strategies. *Expected Attention*, adopts insights from these heuristic approaches while providing a principled theoretical foundation based on the distributional properties of transformer activations.

**Quantization** Instead of reducing the KV cache size along the sequence dimension, quantization methods try to reduce the precision used to store the cache. For example, NQKV Cai et al. (2025b) partitions the cache into blocks for quantization and processes them separately. KVQuant (Hooper et al., 2024) performs non uniform per-layer quantization, while KIVI (Zirui Liu et al., 2023) quantizes the key cache by layer and the value cache by token. These methods are orthogonal to Expected Attention (and to KV cache compression in general), making it possible to integrate them.

**Efficient Implementations** Alongside compression, sparse attention and quantization, another effort has been done to devise efficient implementation of inference systems. In this context, a well designed low-level handling of the KV cache can deliver significant performance speed-ups, especially in multi-user serving systems. The first to investigate this and introduce efficient memory management for KV cache was vLLM (Kwon et al., 2023), soon followed by other approaches (Prabhu et al., 2024; Jiang et al., 2024) and frameworks (NVIDIA, 2024).

## 7 LIMITATIONS

A key trade-off of our training-free methodology is that its performance does not match that of trainable methods (DeepSeek-AI, 2024a; Łańcucki et al., 2025). This is an intentional design choice that allows deployment without significant computational resources required for intensive training. Future work could explore combining our theoretical framework with lightweight fine-tuning.

Another limitation is that our method requires users to specify compression ratios manually, lacking an automated mechanism to determine optimal compression levels for different scenarios such as text generation. This represents a promising area for future research.

Finally, while our PyTorch implementation effectively demonstrates our method's theoretical principles, it is not optimized for efficiency. A highly performant implementation with custom CUDA kernels would significantly improve speed and practical utility.

## 8 CONCLUSION

We introduced Expected Attention, a training-free algorithm for KV cache compression. We showed Expected Attention outperforms state-of-art KV cache compression methods on several benchmarks and in both prefilling and decoding scenarios. Additionally, we released a research library that allows researchers to easily implement and experiment with KV cache compression methods, and evaluate them on popular benchmarks for long context.

## 9 REPRODUCIBILITY STATEMENT

To ensure the reproducibility of our work, we are providing a complete and self-contained codebase along with this submission. The provided code includes all necessary scripts for data preprocessing and evaluation, allowing for the direct replication of our experiments and results. For now, we share the repo in an anonymized github repository.

The codebase is organized to be straightforward to use and is accompanied by a `README.md` file with detailed instructions on how to set up the environment and run the experiments.

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

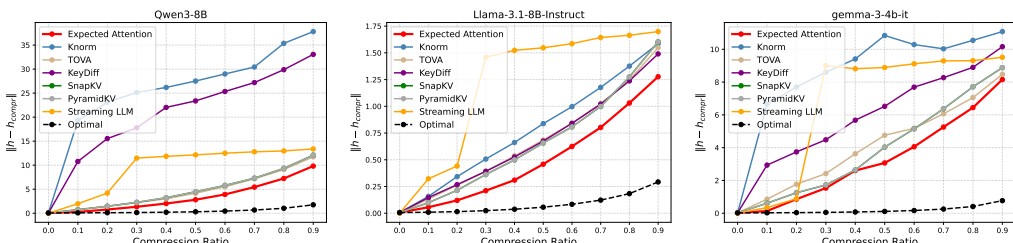

Figure 6: Reconstruction error $\|h - h_{\text{compr}}\|$ averaged across model layers. Expected Attention achieves the best error, minimizing the impact on the residual stream.

## A    RECONSTRUCTION ERROR ACROSS METHODS

In Section 2, we discussed the challenge of compressing the KV cache without significantly altering the residual stream. To understand the impact of Expected Attention on the model output, we quantify the reconstruction error of the residual stream, i.e. how the difference between the original, uncompressed hidden states and the corresponding hidden states after compression. We define the reconstruction error as $\|h - h_{\text{compr}}\|$, where $h$ is the original hidden state without compression and $h_{\text{compr}}$ the hidden state after the KV cache has been compressed. We average the reconstrcution error over a long sequence of $\sim$ 5K tokens and display the results for several methods in Figure 6. Expected Attention consistently achieves a lower reconstruction error, indicating that it preserves the integrity of the hidden state more effectively than competing methods, a crucial property for maintaining downstream performance (Mudarisov et al., 2025; Gordić, 2025).

## B    DISTRIBUTIONAL PROPERTIES OF LLM ACTIVATIONS

In this section, we analyse the distributional properties of activations within Large Language Models. Our investigation aligns with the findings of prior work, which has demonstrated that LLM activations often exhibit normal distributions. More specifically Liu et al. (2025) finds that hidden states are zero-mean unimodal, and qualitatively fall into two distinctly shaped distributions. The hidden states before the Attention and the MLP layers tend to be Gaussian-like, while the hidden states in the intermediate of such layers tend to be Laplacian-like.

For Expected Attention, we are interested in the hidden states before the MLP layers and the corresponding queries. Our study confirms that such activations are predominantly unimodal and can be approximated as Gaussian distributions, albeit with the presence of a few heavy-tailed outliers, as already found in Xiao et al. (2023); Sun et al. (2024). Importantly, EA does not require strict Gaussianity, the essential property is unimodality. In Figure 9a, Figure 8a, and Figure 7a we show hidden states and queries for different models. For our method, the distributional properties of queries are of particular importance, and we observe that queries maintain a clear Gaussian-like behaviour. This also applies to models with QK normalization, where the query projection is not guaranteed to be linear. The concentration of these activations around a central value and their Gaussian like shape provides the theoretical basis for Expected Attention.

We stress that in this work, our goal is not to explain or investigate this property, but rather to leverage it for KV cache compression.

## C    EXPECTED ATTENTION SCORE

To empirically validate that the expected attention score is strongly correlated to the real model attention score, we plot the correlation between the observed attention and the expected attention score across different layers and heads. We use sequence of 5K tokens and use the first 1K tokens to compute the query statistics. We display the results in Figure 10. We see that for different layers and attention heads, the expected attention score from Equation 4 is strongly correlated to the original attention score.

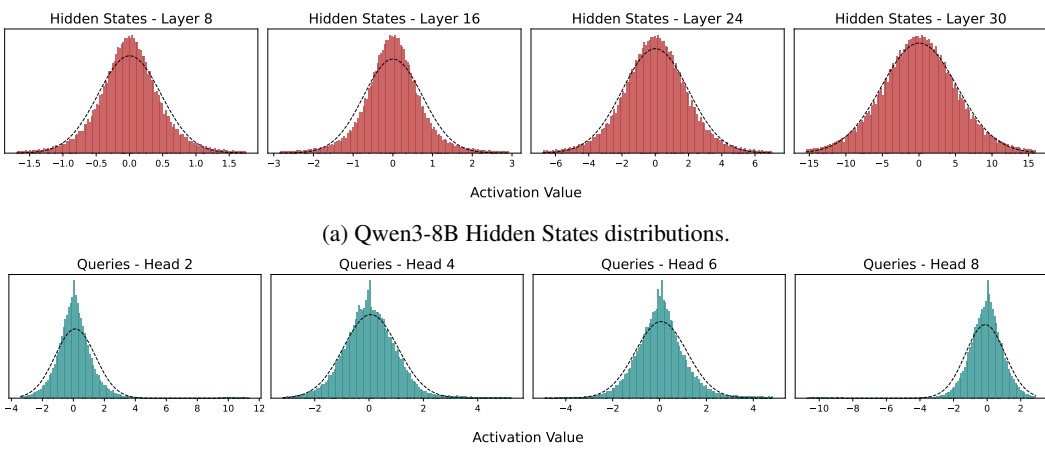

(a) Qwen3-8B Hidden States distributions.

(b) Qwen3-8B queries distributions.

Figure 7: Distributions of Qwen3-8B Hidden States and queries.

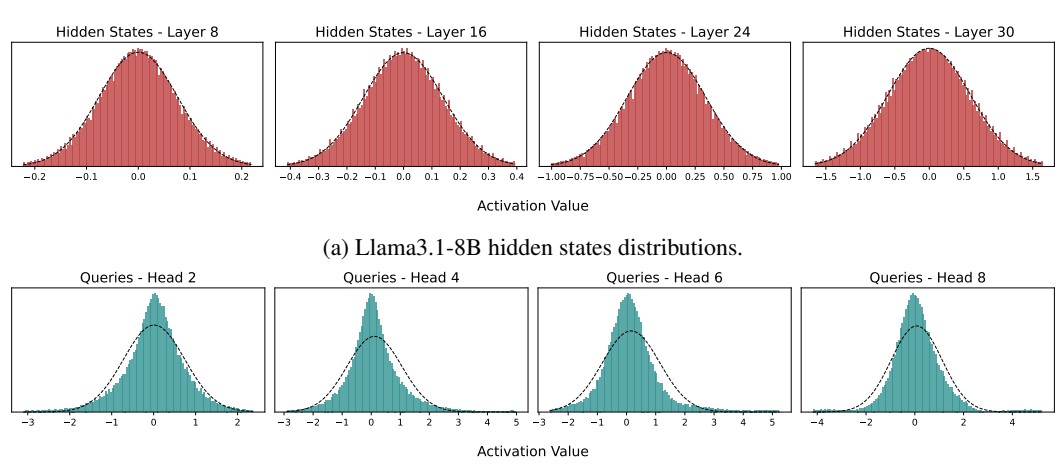

(a) Llama3.1-8B hidden states distributions.

(b) Llama3.1-8B queries distributions.

Figure 8: Distributions of Llama3.1-8B hidden states and queries.

## D    CONTRIBUTION OF NORM OF THE VALUES

We perform an additional ablation study on the impact of using the norm of values to contribute to the Expected Attention score. The results, summarized in Table 5, clearly demonstrate the substantial importance of incorporating the norm of values into the score calculation. Across all tested models, removing the value norm contribution leads to a drastic reduction in performance, confirming that the magnitude of the value vectors plays a critical role in determining the overall attention outcome. This ablation strongly confirms the findings presented in Guo et al. (2024).

Table 5: Ablation results showing the performance comparison on Ruler 4K of the Expected Attention (EA) method with and without the contribution of the Value Norm.

| Model | EA | w/o Value Norm |
|---|---|---|
| **Llama3.1-8B** | 92.2 | 77.7 |
| **Qwen3-8B** | 94.7 | 48.9 |
| **Gemma3-12B** | 92.7 | 44.9 |

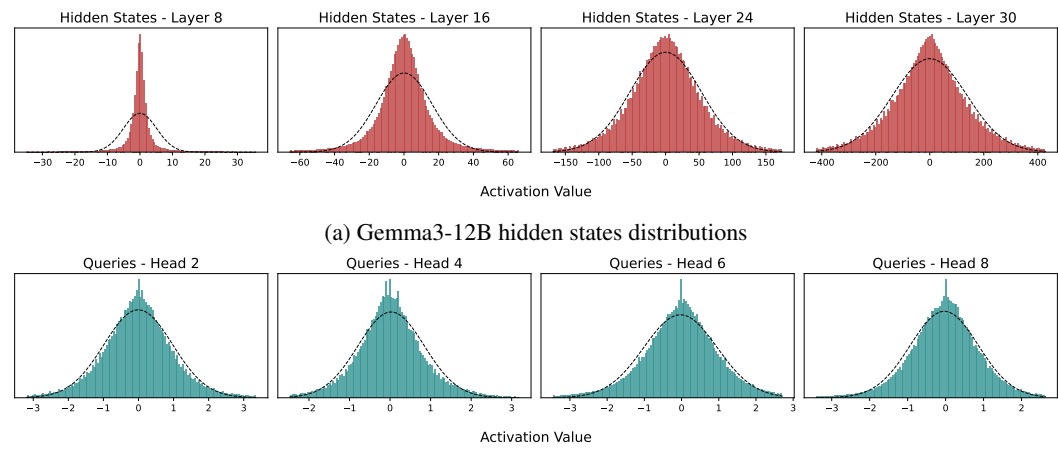

(a) Gemma3-12B hidden states distributions

(b) Gemma3-12B queries distributions.

Figure 9: Distributions of Gemma3-12B hidden states and queries.

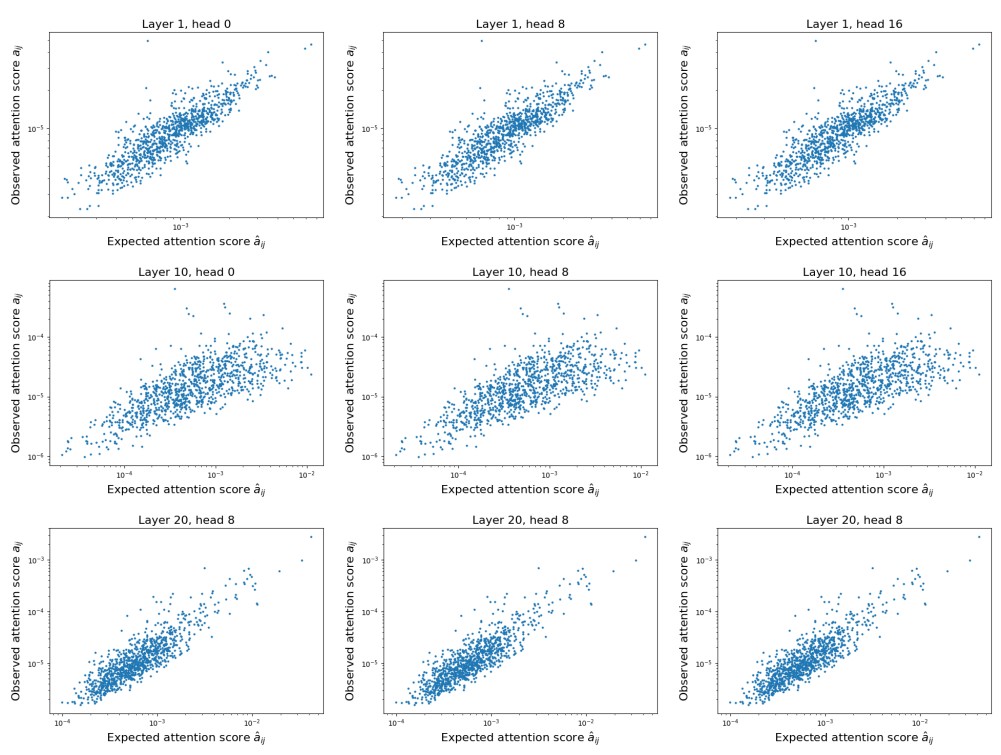

Figure 10: Correlation between attention score and expected attention score for Llama3.1-8B. We compute the expected attentions score on a sequence of 5K tokens, using the first 1K for statistics. A strong correlation exists between our attention score approximation and the observed attention score.

## E  ADDITIONAL RESULTS

In Table 6 we show additional results on the LongBench dataset, averaged across all subsets. The results for **Gemma3-12B** on LongBench exhibit behaviors that differ from other models. Specifically, all compression methods show an initial increase in average score at the $10\%$ and $25\%$ compression ratios compared to the $0\%$ baseline. This unexpected gain suggests that removing a small fraction of the least-important Key-Value pairs effectively prunes noisy or redundant information,

Table 6: Expected Attention outperforms most baselines on Longbench (Bai et al., 2024). We show average score with increasing compression ratios across baselines.

| Model | Method | Longbench | | | | | |
|---|---|---|---|---|---|---|---|
| | | 0% | 10% | 25% | 50% | 75% | 90% |
| *Qwen3-8B* | Expected Attention | **48.63** | 48.30 | **50.25** | **50.1** | **48.06** | **39.71** |
| | TOVA | **48.63** | 48.41 | 48.14 | 46.49 | 43.19 | 37.21 |
| | SnapKV | **48.63** | **48.40** | 47.85 | 46.25 | 42.42 | 34.57 |
| | KeyDiff | **48.63** | 48.13 | 46.23 | 40.08 | 29.42 | 20.69 |
| *Gemma3-12B* | Expected Attention | **51.04** | **54.02** | 50.98 | 47.51 | 40.41 | 32.67 |
| | TOVA | **51.04** | 53.05 | **51.52** | **50.7** | **46.88** | **40.45** |
| | SnapKV | **51.04** | 51.83 | 51.31 | 48.14 | 44.31 | 34.97 |
| | KeyDiff | **51.04** | 51.64 | 48.74 | 42.15 | 33.68 | 23.46 |
| *Llama3.1-8B* | Expected Attention | **46.42** | **46.59** | 46.8 | **47.91** | 44.04 | 33.97 |
| | TOVA | **46.42** | 46.22 | 45.62 | 44.13 | 40.5 | 34.77 |
| | SnapKV | **46.42** | 46.56 | 46.07 | 45.07 | 41.24 | 32.55 |
| | KeyDiff | **46.42** | 46.45 | 48.01 | 46.9 | 42.24 | **35.51** |

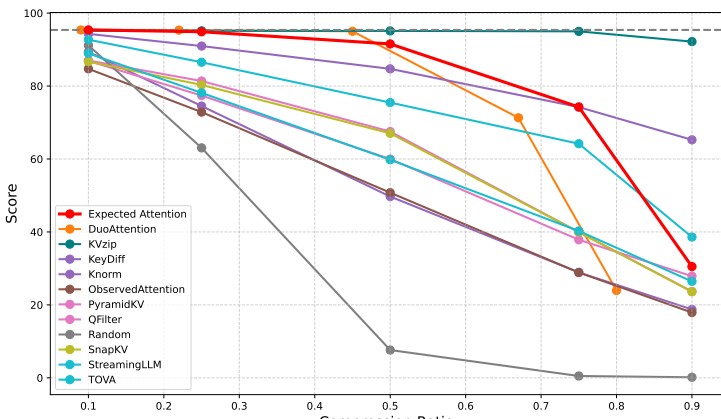

Figure 11: Initial experiments on Ruler 4K to select the best baselines. We did not use KVZip as it requires two forward passes and increases latency significantly.

thereby enhancing performance. However, at higher compression ratios (50%, 75%, and 90%), the attention-based method TOVA maintains superior scores compared to Expected Attention (EA) and other baselines.

**Ruler** In order to select the most competitive baselines we performed an initial search on 15+ methods on Ruler. We selected the best performing ones as displayed in Figure 11. We did not include KVZip (Kim et al., 2025) despite achieving a high score as it needs two forward passes, therefore implying a higher cost FLOPs that is double as much as the other baselines.

# F   FLOPs CALCULATION

We follow Hoffmann et al. (2022) and include all inference FLOPs, including those contributed to by the embedding matrices, in our analysis. Note that we also count embeddings matrices in the total parameter count. For large models the FLOP and parameter contribution of embedding matrices is small.

For the forward pass, we consider contributions from:

- Embeddings
    - $2 \times$ seq_len $\times$ vocab_size $\times$ d_model
- Attention (Single Layer)
    - **Key, query and value projections**: $2 \times 3 \times$ seq_len $\times$ d_model $\times$ (key_size $\times$ num_heads)
    - **Key @ Query logits**: $2 \times$ seq_len $\times$ seq_len $\times$ (key_size $\times$ num_heads)
    - **Softmax**: $3 \times$ num_heads $\times$ seq_len $\times$ seq_len
    - **Softmax @ query reductions**: $2 \times$ seq_len $\times$ seq_len $\times$ (key_size $\times$ num_heads)
    - **Final Linear**: $2 \times$ seq_len $\times$ (key_size $\times$ num_heads) $\times$ d_model
- Dense Block (Single Layer)
    - $2 \times$ seq_len $\times$ (d_model $\times$ ffw_size + d_model $\times$ ffw_size)
- Final Logits
    - $2 \times$ seq_len $\times$ d_model $\times$ vocab_size
- **Total forward pass FLOPs:** embeddings + num_layers $\times$ (total_attention + dense_block) + logits

# G   DERIVATION FOR EXPECTED ATTENTION SCORE

The equality in Equation eq. (7) is derived by applying the formula for the expected value of the exponential of a Gaussian random variable.

- The term in the exponent, $X = \frac{\mathbf{q}^T \mathbf{k}_i}{\sqrt{d}}$, is a Gaussian random variable, $X \sim \mathcal{N}(\mu_X, \sigma_X^2)$.
- The mean of $X$ is $\mu_X = \mathbb{E}[X] = \frac{\bar{\mu}_q^T \mathbf{k}_i}{\sqrt{d}}$.
- The variance of $X$ is $\sigma_X^2 = \mathrm{Var}(X) = \frac{\mathbf{k}_i^T \bar{\Sigma}_q \mathbf{k}_i}{d}$.

The expectation $\mathbb{E}[\exp(X)]$ is then computed using the Moment-Generating Function (MGF) $M_X(t) = \mathbb{E}[e^{tX}]$ of a Gaussian distribution, evaluated at $t = 1$. Since $M_X(t) = \exp(\mu_X t + \frac{1}{2}\sigma_X^2 t^2)$, setting $t = 1$ yields the identity:

$$\mathbb{E}[\exp(X)] = \exp\left(\mu_X + \frac{\sigma_X^2}{2}\right)$$

Substituting $\mu_X$ and $\sigma_X^2$ recovers Equation eq. (7).

# H   DETAILED RESULTS ON RULER

# I   LLM USAGE STATEMENT

We used LLMs to polish the text and refine the language.

Table 7: **Common Word Extraction**

| Model | Method | Ruler 4k | | | | | | Ruler 16k | | | | | |
|---|---|---|---|---|---|---|---|---|---|---|---|---|---|
| | | 0% | 10% | 25% | 50% | 75% | 90% | 0% | 10% | 25% | 50% | 75% | 90% |
| Qwen3-8B | EA (Ours) | **98.9** | 98.9 | 98.5 | 96.1 | 44.9 | 12.6 | **82.8** | 85.2 | 86.8 | 87.8 | 84.4 | 35.2 |
| | TOVA[51] | **98.9** | 98.7 | 96.4 | 84.2 | 51.8 | 15.4 | **82.8** | 83.7 | 83.5 | 77.5 | 57.8 | 20.2 |
| | SnapKV[38] | **98.9** | 98.9 | 99.0 | 98.5 | 92.6 | 49.2 | **82.8** | 83.4 | 82.6 | 78.2 | 58.7 | 19.2 |
| | KeyDiff[52] | **98.9** | 98.0 | 97.1 | 90.7 | 66.2 | 6.9 | **82.8** | 84.3 | 85.2 | 86.4 | 82.8 | 64.7 |
| Gemma3-12B | EA (Ours) | **95.0** | 95.0 | 95.3 | 97.8 | 94.8 | 65.7 | **89.8** | 87.1 | 86.6 | 87.1 | 78.1 | 23.2 |
| | TOVA[51] | **95.0** | 94.9 | 94.8 | 94.8 | 90.7 | 53.8 | **89.8** | 89.8 | 89.9 | 90.5 | 89.5 | 78.1 |
| | SnapKV[38] | **95.0** | 95.8 | 96.5 | 96.3 | 94.8 | 81.1 | **89.8** | 90.0 | 90.0 | 90.3 | 88.6 | 73.8 |
| | KeyDiff[52] | **95.0** | 95.3 | 95.5 | 84.6 | 35.2 | 9.1 | **89.8** | 89.2 | 87.7 | 84.5 | 42.5 | 12.1 |
| Llama3.1-8B | EA (Ours) | **99.6** | 99.7 | 99.6 | 99.4 | 92.7 | 51.8 | **89.5** | 89.2 | 86.9 | 81.9 | 28.6 | 2.2 |
| | TOVA[51] | **99.6** | 99.3 | 97.2 | 85.0 | 52.8 | 23.9 | **89.5** | 89.1 | 90.1 | 91.3 | 85.8 | 60.5 |
| | SnapKV[38] | **99.6** | 99.7 | 99.5 | 97.4 | 84.4 | 38.9 | **89.5** | 88.1 | 85.5 | 71.7 | 17.8 | 0.3 |
| | KeyDiff[52] | **99.6** | 99.5 | 99.1 | 94.3 | 56.9 | 10.7 | **89.5** | 89.4 | 88.9 | 87.2 | 71.1 | 25.8 |

Table 8: **Frequent Words Extraction**

| Model | Method | Ruler 4k | | | | | | Ruler 16k | | | | | |
|---|---|---|---|---|---|---|---|---|---|---|---|---|---|
| | | 0% | 10% | 25% | 50% | 75% | 90% | 0% | 10% | 25% | 50% | 75% | 90% |
| Qwen3-8B | EA (Ours) | **95.3** | 95.3 | 95.5 | 96.1 | 91.4 | 56.4 | **93.9** | 93.4 | 93.4 | 92.9 | 92.1 | 87.1 |
| | TOVA[51] | **95.3** | 94.8 | 93.0 | 89.8 | 81.1 | 59.4 | **93.9** | 94.4 | 95.5 | 96.5 | 97.6 | 97.0 |
| | SnapKV[38] | **95.3** | 96.1 | 95.4 | 93.8 | 88.3 | 77.4 | **93.9** | 94.3 | 94.7 | 95.2 | 93.9 | 91.4 |
| | KeyDiff[52] | **95.3** | 93.7 | 91.3 | 85.1 | 68.4 | 36.9 | **93.9** | 94.8 | 94.9 | 94.5 | 88.8 | 65.9 |
| Gemma3-12B | EA (Ours) | **97.3** | 97.3 | 97.3 | 97.1 | 91.8 | 69.7 | **98.6** | 98.0 | 97.8 | 97.2 | 94.5 | 86.4 |
| | TOVA[51] | **97.3** | 97.4 | 97.0 | 94.4 | 85.8 | 64.8 | **98.6** | 98.7 | 98.9 | 98.7 | 97.9 | 93.3 |
| | SnapKV[38] | **97.3** | 97.5 | 97.3 | 97.1 | 93.7 | 86.2 | **98.6** | 99.0 | 99.1 | 98.8 | 98.0 | 96.0 |
| | KeyDiff[52] | **97.3** | 97.2 | 96.2 | 90.5 | 78.2 | 57.0 | **98.6** | 98.7 | 97.1 | 94.0 | 87.6 | 62.9 |
| Llama3.1-8B | EA (Ours) | **94.8** | 94.8 | 94.5 | 96.0 | 91.5 | 52.3 | **90.1** | 90.0 | 89.8 | 88.1 | 84.0 | 28.7 |
| | TOVA[51] | **94.8** | 93.5 | 90.7 | 84.0 | 70.5 | 30.6 | **90.1** | 90.5 | 90.8 | 90.7 | 87.9 | 76.2 |
| | SnapKV[38] | **94.8** | 94.8 | 94.2 | 89.8 | 85.6 | 61.0 | **90.1** | 90.5 | 91.5 | 88.4 | 77.0 | 62.1 |
| | KeyDiff[52] | **94.8** | 94.9 | 94.7 | 92.9 | 85.8 | 70.1 | **90.1** | 89.9 | 89.3 | 88.9 | 87.5 | 84.3 |

Table 9: **NIAH Multikey 1**

| Model | Method | Ruler 4k | | | | | | Ruler 16k | | | | | |
|---|---|---|---|---|---|---|---|---|---|---|---|---|---|
| | | 0% | 10% | 25% | 50% | 75% | 90% | 0% | 10% | 25% | 50% | 75% | 90% |
| Qwen3-8B | EA (Ours) | **100.0** | 90.8 | 66.2 | 25.4 | 0.2 | 0.0 | **99.6** | 99.8 | 99.2 | 98.8 | 98.2 | 83.8 |
| | TOVA[51] | **100.0** | 100.0 | 100.0 | 100.0 | 94.8 | 28.2 | **99.6** | 99.6 | 99.6 | 99.6 | 99.4 | 86.2 |
| | SnapKV[38] | **100.0** | 98.0 | 84.6 | 39.6 | 19.2 | 12.0 | **99.6** | 99.4 | 97.4 | 68.4 | 24.6 | 12.6 |
| | KeyDiff[52] | **100.0** | 98.6 | 97.0 | 94.8 | 79.0 | 52.2 | **99.6** | 96.4 | 87.0 | 82.4 | 70.6 | 52.4 |
| Gemma3-12B | EA (Ours) | **99.6** | 100.0 | 99.8 | 98.8 | 85.2 | 50.8 | **90.4** | 86.2 | 83.8 | 79.2 | 57.4 | 30.6 |
| | TOVA[51] | **99.6** | 99.8 | 99.6 | 97.8 | 64.0 | 9.8 | **90.4** | 89.4 | 88.0 | 71.0 | 35.0 | 8.2 |
| | SnapKV[38] | **99.6** | 82.2 | 60.2 | 27.2 | 15.4 | 10.8 | **90.4** | 78.8 | 57.6 | 25.6 | 12.2 | 10.4 |
| | KeyDiff[52] | **99.6** | 99.2 | 99.2 | 97.4 | 81.8 | 38.8 | **90.4** | 79.8 | 78.0 | 76.2 | 60.8 | 34.2 |
| Llama3.1-8B | EA (Ours) | **99.8** | 99.8 | 99.6 | 94.8 | 61.2 | 10.2 | **99.8** | 100.0 | 100.0 | 99.6 | 95.6 | 15.6 |
| | TOVA[51] | **99.8** | 99.8 | 99.8 | 99.8 | 98.2 | 64.8 | **99.8** | 99.6 | 99.6 | 99.6 | 99.6 | 94.6 |
| | SnapKV[38] | **99.8** | 99.8 | 98.8 | 99.8 | 34.4 | 99.4 | **99.8** | 99.4 | 99.6 | 95.2 | 52.4 | 17.4 |
| | KeyDiff[52] | **99.8** | 99.8 | 100.0 | 100.0 | 100.0 | 97.6 | **99.8** | 99.6 | 99.6 | 99.4 | 99.6 | 99.4 |

Table 10: **NIAH Multikey 2**

| Model | Method | Ruler 4k | | | | | | Ruler 16k | | | | | |
|---|---|---|---|---|---|---|---|---|---|---|---|---|---|
| | | 0% | 10% | 25% | 50% | 75% | 90% | 0% | 10% | 25% | 50% | 75% | 90% |
| *Qwen3-8B* | EA (Ours) | **100.0** | 100.0 | 97.6 | 47.8 | 6.8 | 0.2 | **100.0** | 100.0 | 99.6 | 99.6 | 96.8 | 21.8 |
| | TOVA[51] | **100.0** | 69.0 | 30.4 | 6.6 | 1.4 | 0.2 | **100.0** | 76.2 | 35.6 | 9.4 | 1.0 | 0.4 |
| | SnapKV[38] | **100.0** | 92.2 | 75.2 | 30.6 | 9.8 | 2.8 | **100.0** | 93.2 | 74.8 | 33.8 | 7.8 | 1.4 |
| | KeyDiff[52] | **100.0** | 99.0 | 89.8 | 55.0 | 13.0 | 1.2 | **100.0** | 87.4 | 73.4 | 32.0 | 3.8 | 0.2 |
| *Gemma3-12B* | EA (Ours) | **98.8** | 98.6 | 98.8 | 95.6 | 71.4 | 4.6 | **55.4** | 53.6 | 53.2 | 41.6 | 15.8 | 3.2 |
| | TOVA[51] | **98.8** | 60.2 | 10.4 | 1.0 | 0.0 | 0.0 | **55.4** | 30.8 | 8.6 | 1.4 | 0.0 | 0.0 |
| | SnapKV[38] | **98.8** | 97.6 | 89.0 | 51.8 | 14.2 | 3.4 | **55.4** | 55.0 | 44.0 | 21.6 | 5.4 | 1.6 |
| | KeyDiff[52] | **98.8** | 96.6 | 93.0 | 64.0 | 13.4 | 1.0 | **55.4** | 46.4 | 40.0 | 24.6 | 6.0 | 1.0 |
| *Llama3.1-8B* | EA (Ours) | **100.0** | 100.0 | 99.6 | 88.2 | 30.8 | 2.2 | **100.0** | 100.0 | 99.6 | 95.8 | 66.0 | 2.8 |
| | TOVA[51] | **100.0** | 96.0 | 72.8 | 33.2 | 7.0 | 2.6 | **100.0** | 90.4 | 67.4 | 28.4 | 8.8 | 1.8 |
| | SnapKV[38] | **100.0** | 100.0 | 84.2 | 99.4 | 17.2 | 84.4 | **100.0** | 96.6 | 80.8 | 45.0 | 18.2 | 3.8 |
| | KeyDiff[52] | **100.0** | 99.8 | 98.8 | 88.8 | 30.0 | 2.8 | **100.0** | 100.0 | 98.0 | 76.0 | 27.2 | 3.0 |

Table 11: **NIAH Multikey 3**

| Model | Method | Ruler 4k | | | | | | Ruler 16k | | | | | |
|---|---|---|---|---|---|---|---|---|---|---|---|---|---|
| | | 0% | 10% | 25% | 50% | 75% | 90% | 0% | 10% | 25% | 50% | 75% | 90% |
| *Qwen3-8B* | EA (Ours) | **100.0** | 87.2 | 47.2 | 9.8 | 0.2 | 0.0 | **99.6** | 99.8 | 99.8 | 99.0 | 45.4 | 0.0 |
| | TOVA[51] | **100.0** | 50.4 | 12.2 | 0.4 | 0.0 | 0.0 | **99.6** | 63.8 | 24.4 | 4.0 | 0.6 | 0.0 |
| | SnapKV[38] | **100.0** | 87.8 | 58.4 | 20.0 | 1.6 | 0.0 | **99.6** | 88.8 | 59.6 | 18.0 | 3.8 | 1.0 |
| | KeyDiff[52] | **100.0** | 92.8 | 67.8 | 18.6 | 2.0 | 0.0 | **99.6** | 82.8 | 50.2 | 14.2 | 0.6 | 0.0 |
| *Gemma3-12B* | EA (Ours) | **99.8** | 99.6 | 99.2 | 86.8 | 11.8 | 0.0 | **61.6** | 45.8 | 41.4 | 25.0 | 8.8 | 0.0 |
| | TOVA[51] | **99.8** | 65.8 | 8.8 | 0.0 | 0.0 | 0.0 | **61.6** | 26.8 | 8.8 | 0.6 | 0.0 | 0.0 |
| | SnapKV[38] | **99.8** | 93.2 | 68.2 | 28.8 | 2.4 | 0.0 | **61.6** | 56.2 | 41.2 | 9.6 | 2.0 | 0.6 |
| | KeyDiff[52] | **99.8** | 92.0 | 68.2 | 7.2 | 0.0 | 0.0 | **61.6** | 32.4 | 21.4 | 10.6 | 0.0 | 0.0 |
| *Llama3.1-8B* | EA (Ours) | **99.8** | 100.0 | 99.8 | 27.0 | 0.2 | 0.0 | **99.2** | 99.2 | 99.0 | 54.6 | 10.8 | 0.0 |
| | TOVA[51] | **99.8** | 74.6 | 33.4 | 2.6 | 0.0 | 0.0 | **99.2** | 77.2 | 39.2 | 8.2 | 1.0 | 0.4 |
| | SnapKV[38] | **99.8** | 99.8 | 55.2 | 84.0 | 1.6 | 0.0 | **99.2** | 86.6 | 60.2 | 18.2 | 3.6 | 1.0 |
| | KeyDiff[52] | **99.8** | 87.2 | 53.2 | 11.0 | 0.0 | 0.0 | **99.2** | 82.8 | 43.2 | 6.8 | 0.0 | 0.0 |

Table 12: **NIAH Multiquery**

| Model | Method | Ruler 4k | | | | | | Ruler 16k | | | | | |
|---|---|---|---|---|---|---|---|---|---|---|---|---|---|
| | | 0% | 10% | 25% | 50% | 75% | 90% | 0% | 10% | 25% | 50% | 75% | 90% |
| *Qwen3-8B* | EA (Ours) | **99.9** | 93.7 | 76.4 | 25.4 | 0.1 | 0.0 | **100.0** | 100.0 | 99.8 | 99.6 | 99.6 | 94.1 |
| | TOVA[51] | **99.9** | 99.9 | 99.9 | 100.0 | 96.7 | 21.4 | **100.0** | 100.0 | 100.0 | 99.9 | 100.0 | 85.0 |
| | SnapKV[38] | **99.9** | 99.3 | 88.4 | 40.9 | 16.9 | 10.8 | **100.0** | 100.0 | 97.1 | 67.7 | 20.6 | 10.7 |
| | KeyDiff[52] | **99.9** | 100.0 | 99.8 | 99.2 | 92.7 | 65.3 | **100.0** | 99.8 | 98.6 | 97.8 | 94.3 | 79.5 |
| *Gemma3-12B* | EA (Ours) | **100.0** | 100.0 | 99.9 | 99.8 | 88.7 | 63.0 | **99.2** | 99.1 | 98.8 | 98.5 | 83.2 | 41.0 |
| | TOVA[51] | **100.0** | 100.0 | 100.0 | 98.3 | 60.0 | 1.6 | **99.2** | 98.8 | 95.8 | 81.0 | 36.1 | 6.2 |
| | SnapKV[38] | **100.0** | 86.2 | 56.5 | 20.6 | 11.3 | 9.9 | **99.2** | 88.3 | 63.5 | 23.1 | 11.2 | 10.3 |
| | KeyDiff[52] | **100.0** | 100.0 | 100.0 | 99.3 | 89.2 | 42.1 | **99.2** | 99.2 | 99.2 | 98.9 | 91.5 | 58.0 |
| *Llama3.1-8B* | EA (Ours) | **99.9** | 98.5 | 99.8 | 80.8 | 44.1 | 4.6 | **99.0** | 99.0 | 99.0 | 99.0 | 97.0 | 13.3 |
| | TOVA[51] | **99.9** | 99.9 | 99.9 | 99.9 | 97.5 | 50.8 | **99.0** | 99.0 | 99.1 | 99.3 | 99.3 | 94.7 |
| | SnapKV[38] | **99.9** | 99.9 | 95.7 | 99.9 | 26.6 | 92.0 | **99.0** | 99.0 | 98.7 | 84.0 | 34.9 | 13.6 |
| | KeyDiff[52] | **99.9** | 99.9 | 99.9 | 100.0 | 99.8 | 98.9 | **99.0** | 99.0 | 99.1 | 99.2 | 99.5 | 99.4 |

Table 13: **NIAH Multivalue**

| Model | Method | Ruler 4k | | | | | | Ruler 16k | | | | | |
|---|---|---|---|---|---|---|---|---|---|---|---|---|---|
| | | 0% | 10% | 25% | 50% | 75% | 90% | 0% | 10% | 25% | 50% | 75% | 90% |
| Qwen3-8B | EA (Ours) | **100.0** | 98.0 | 83.7 | 30.3 | 0.3 | 0.0 | **99.6** | 99.5 | 99.6 | 99.5 | 99.2 | 93.9 |
| | TOVA[51] | **100.0** | 100.0 | 100.0 | 99.9 | 96.9 | 22.2 | **99.6** | 99.5 | 99.6 | 99.7 | 99.1 | 82.7 |
| | SnapKV[38] | **100.0** | 99.0 | 89.0 | 39.6 | 12.7 | 10.0 | **99.6** | 99.6 | 96.5 | 64.2 | 17.1 | 9.8 |
| | KeyDiff[52] | **100.0** | 100.0 | 100.0 | 99.8 | 94.2 | 57.6 | **99.6** | 99.3 | 98.6 | 98.8 | 97.6 | 78.7 |
| Gemma3-12B | EA (Ours) | **99.7** | 99.5 | 98.5 | 95.3 | 86.4 | 68.1 | **95.5** | 89.0 | 84.5 | 74.2 | 65.5 | 39.4 |
| | TOVA[51] | **99.7** | 99.7 | 99.7 | 98.6 | 56.5 | 1.6 | **95.5** | 95.2 | 91.0 | 72.2 | 27.4 | 4.3 |
| | SnapKV[38] | **99.7** | 80.5 | 43.9 | 16.8 | 10.6 | 9.7 | **95.5** | 79.1 | 45.4 | 13.4 | 10.1 | 9.8 |
| | KeyDiff[52] | **99.7** | 99.8 | 99.8 | 98.5 | 87.7 | 35.3 | **95.5** | 95.5 | 95.2 | 94.7 | 89.0 | 53.6 |
| Llama3.1-8B | EA (Ours) | **99.9** | 98.0 | 99.7 | 86.1 | 47.2 | 4.8 | **98.9** | 98.7 | 98.5 | 97.6 | 81.1 | 14.8 |
| | TOVA[51] | **99.9** | 99.9 | 99.8 | 99.7 | 96.9 | 51.3 | **98.9** | 99.2 | 99.0 | 99.0 | 99.0 | 92.7 |
| | SnapKV[38] | **99.9** | 99.8 | 90.1 | 99.3 | 25.3 | 51.3 | **98.9** | 98.8 | 96.2 | 79.8 | 30.1 | 12.7 |
| | KeyDiff[52] | **99.9** | 99.8 | 99.9 | 99.8 | 99.0 | 96.2 | **98.9** | 99.2 | 99.0 | 99.1 | 98.8 | 98.7 |

Table 14: **NIAH Single 1**

| Model | Method | Ruler 4k | | | | | | Ruler 16k | | | | | |
|---|---|---|---|---|---|---|---|---|---|---|---|---|---|
| | | 0% | 10% | 25% | 50% | 75% | 90% | 0% | 10% | 25% | 50% | 75% | 90% |
| Qwen3-8B | EA (Ours) | **100.0** | 100.0 | 100.0 | 97.6 | 22.8 | 0.0 | **100.0** | 100.0 | 100.0 | 100.0 | 100.0 | 99.8 |
| | TOVA[51] | **100.0** | 100.0 | 100.0 | 100.0 | 80.8 | 17.4 | **100.0** | 100.0 | 100.0 | 100.0 | 98.2 | 65.2 |
| | SnapKV[38] | **100.0** | 92.8 | 88.2 | 74.6 | 39.2 | 5.2 | **100.0** | 100.0 | 100.0 | 98.6 | 93.0 | 70.6 |
| | KeyDiff[52] | **100.0** | 100.0 | 100.0 | 100.0 | 100.0 | 100.0 | **100.0** | 100.0 | 100.0 | 100.0 | 100.0 | 100.0 |
| Gemma3-12B | EA (Ours) | **100.0** | 100.0 | 100.0 | 100.0 | 100.0 | 99.6 | **100.0** | 100.0 | 100.0 | 100.0 | 100.0 | 100.0 |
| | TOVA[51] | **100.0** | 100.0 | 100.0 | 99.8 | 98.2 | 36.8 | **100.0** | 100.0 | 100.0 | 100.0 | 100.0 | 84.8 |
| | SnapKV[38] | **100.0** | 99.6 | 99.2 | 98.6 | 94.0 | 72.6 | **100.0** | 100.0 | 100.0 | 99.6 | 96.4 | 85.6 |
| | KeyDiff[52] | **100.0** | 100.0 | 100.0 | 100.0 | 99.8 | 81.6 | **100.0** | 100.0 | 100.0 | 100.0 | 100.0 | 100.0 |
| Llama3.1-8B | EA (Ours) | **100.0** | 99.8 | 100.0 | 97.6 | 95.6 | 93.8 | **100.0** | 100.0 | 100.0 | 100.0 | 100.0 | 99.4 |
| | TOVA[51] | **100.0** | 100.0 | 100.0 | 100.0 | 100.0 | 89.8 | **100.0** | 100.0 | 100.0 | 100.0 | 100.0 | 99.6 |
| | SnapKV[38] | **100.0** | 100.0 | 99.4 | 100.0 | 77.2 | 100.0 | **100.0** | 99.8 | 100.0 | 99.6 | 94.2 | 82.4 |
| | KeyDiff[52] | **100.0** | 100.0 | 100.0 | 100.0 | 100.0 | 100.0 | **100.0** | 100.0 | 100.0 | 100.0 | 100.0 | 100.0 |

Table 15: **NIAH Single 2**

| Model | Method | Ruler 4k | | | | | | Ruler 16k | | | | | |
|---|---|---|---|---|---|---|---|---|---|---|---|---|---|
| | | 0% | 10% | 25% | 50% | 75% | 90% | 0% | 10% | 25% | 50% | 75% | 90% |
| Qwen3-8B | EA (Ours) | **100.0** | 99.6 | 92.0 | 31.8 | 2.4 | 0.2 | **100.0** | 100.0 | 100.0 | 100.0 | 99.8 | 96.0 |
| | TOVA[51] | **100.0** | 100.0 | 100.0 | 100.0 | 100.0 | 81.8 | **100.0** | 100.0 | 100.0 | 100.0 | 100.0 | 99.4 |
| | SnapKV[38] | **100.0** | 100.0 | 99.8 | 70.6 | 14.8 | 5.4 | **100.0** | 100.0 | 99.8 | 95.0 | 58.8 | 9.6 |
| | KeyDiff[52] | **100.0** | 100.0 | 100.0 | 99.8 | 92.8 | 54.8 | **100.0** | 100.0 | 100.0 | 100.0 | 97.0 | 69.6 |
| Gemma3-12B | EA (Ours) | **100.0** | 100.0 | 100.0 | 100.0 | 99.8 | 78.0 | **100.0** | 100.0 | 100.0 | 100.0 | 98.2 | 77.8 |
| | TOVA[51] | **100.0** | 100.0 | 100.0 | 100.0 | 98.6 | 63.4 | **100.0** | 100.0 | 98.2 | 89.8 | 52.0 | 10.8 |
| | SnapKV[38] | **100.0** | 95.2 | 87.0 | 60.2 | 20.4 | 5.0 | **100.0** | 90.8 | 71.4 | 41.4 | 11.6 | 2.6 |
| | KeyDiff[52] | **100.0** | 100.0 | 100.0 | 99.4 | 81.0 | 26.0 | **100.0** | 100.0 | 99.6 | 97.8 | 64.2 | 15.8 |
| Llama3.1-8B | EA (Ours) | **100.0** | 100.0 | 99.8 | 94.2 | 78.2 | 38.6 | **100.0** | 100.0 | 100.0 | 100.0 | 99.6 | 21.8 |
| | TOVA[51] | **100.0** | 100.0 | 100.0 | 100.0 | 99.8 | 97.4 | **100.0** | 100.0 | 100.0 | 100.0 | 100.0 | 99.6 |
| | SnapKV[38] | **100.0** | 100.0 | 99.8 | 100.0 | 55.4 | 97.2 | **100.0** | 100.0 | 100.0 | 96.8 | 80.6 | 35.6 |
| | KeyDiff[52] | **100.0** | 100.0 | 100.0 | 100.0 | 100.0 | 100.0 | **100.0** | 100.0 | 100.0 | 100.0 | 100.0 | 100.0 |

Table 16: **NIAH Single 3**

| Model | Method | Ruler 4k | | | | | | Ruler 16k | | | | | |
|---|---|---|---|---|---|---|---|---|---|---|---|---|---|
| | | 0% | 10% | 25% | 50% | 75% | 90% | 0% | 10% | 25% | 50% | 75% | 90% |
| Qwen3-8B | EA (Ours) | **100.0** | 43.8 | 4.2 | 0.0 | 0.0 | 0.0 | **99.8** | 100.0 | 100.0 | 100.0 | 89.6 | 28.8 |
| | TOVA[51] | **100.0** | 100.0 | 100.0 | 97.8 | 15.2 | 0.0 | **99.8** | 99.8 | 99.8 | 99.8 | 64.0 | 2.8 |
| | SnapKV[38] | **100.0** | 99.8 | 84.2 | 14.4 | 5.2 | 2.4 | **99.8** | 87.0 | 45.4 | 8.8 | 2.4 | 2.4 |
| | KeyDiff[52] | **100.0** | 99.8 | 99.8 | 99.0 | 92.8 | 66.0 | **99.8** | 99.8 | 99.8 | 99.0 | 95.6 | 78.4 |
| Gemma3-12B | EA (Ours) | **100.0** | 100.0 | 100.0 | 99.2 | 81.2 | 24.6 | **100.0** | 100.0 | 100.0 | 94.8 | 29.8 | 12.2 |
| | TOVA[51] | **100.0** | 100.0 | 100.0 | 84.0 | 10.8 | 0.0 | **100.0** | 80.8 | 42.4 | 5.4 | 2.4 | 2.0 |
| | SnapKV[38] | **100.0** | 5.4 | 2.8 | 2.4 | 2.4 | 2.4 | **100.0** | 5.6 | 2.8 | 2.4 | 2.4 | 2.4 |
| | KeyDiff[52] | **100.0** | 100.0 | 99.6 | 99.8 | 91.6 | 48.0 | **100.0** | 100.0 | 100.0 | 100.0 | 96.0 | 45.6 |
| Llama3.1-8B | EA (Ours) | **100.0** | 64.6 | 99.6 | 4.0 | 0.2 | 0.0 | **100.0** | 100.0 | 100.0 | 77.4 | 10.2 | 0.0 |
| | TOVA[51] | **100.0** | 99.8 | 95.4 | 52.2 | 4.6 | 0.2 | **100.0** | 100.0 | 99.2 | 83.4 | 29.0 | 2.6 |
| | SnapKV[38] | **100.0** | 99.0 | 11.2 | 36.8 | 2.4 | 2.4 | **100.0** | 74.6 | 35.8 | 14.4 | 3.2 | 2.4 |
| | KeyDiff[52] | **100.0** | 99.8 | 100.0 | 100.0 | 100.0 | 99.8 | **100.0** | 100.0 | 100.0 | 100.0 | 100.0 | 100.0 |

Table 17: **Question Answering 1**

| Model | Method | Ruler 4k | | | | | | Ruler 16k | | | | | |
|---|---|---|---|---|---|---|---|---|---|---|---|---|---|
| | | 0% | 10% | 25% | 50% | 75% | 90% | 0% | 10% | 25% | 50% | 75% | 90% |
| Qwen3-8B | EA (Ours) | **81.6** | 81.0 | 79.6 | 71.6 | 58.4 | 41.2 | **74.8** | 74.4 | 75.8 | 71.8 | 58.6 | 39.6 |
| | TOVA[51] | **81.6** | 81.6 | 80.4 | 75.6 | 55.8 | 33.6 | **74.8** | 74.0 | 68.8 | 54.2 | 36.6 | 27.4 |
| | SnapKV[38] | **81.6** | 79.8 | 78.0 | 69.0 | 54.6 | 38.8 | **74.8** | 70.0 | 60.2 | 44.4 | 33.0 | 26.6 |
| | KeyDiff[52] | **81.6** | 80.8 | 73.6 | 51.6 | 24.4 | 8.8 | **74.8** | 60.4 | 47.6 | 32.2 | 19.8 | 8.2 |
| Gemma3-12B | EA (Ours) | **87.4** | 88.0 | 85.8 | 80.0 | 65.0 | 45.4 | **76.6** | 71.2 | 69.6 | 59.8 | 37.2 | 22.4 |
| | TOVA[51] | **87.4** | 87.4 | 85.0 | 72.0 | 51.6 | 34.4 | **76.6** | 75.4 | 73.2 | 58.8 | 36.0 | 22.0 |
| | SnapKV[38] | **87.4** | 86.4 | 82.6 | 70.6 | 51.0 | 32.4 | **76.6** | 71.8 | 60.2 | 43.2 | 26.2 | 16.8 |
| | KeyDiff[52] | **87.4** | 87.2 | 74.8 | 58.2 | 29.4 | 13.0 | **76.6** | 73.0 | 60.6 | 33.6 | 12.2 | 5.0 |
| Llama3.1-8B | EA (Ours) | **87.8** | 87.2 | 85.4 | 81.0 | 66.2 | 41.4 | **81.2** | 81.0 | 78.8 | 70.6 | 51.2 | 26.8 |
| | TOVA[51] | **87.8** | 87.8 | 86.8 | 80.2 | 56.4 | 27.0 | **81.2** | 81.2 | 80.0 | 64.8 | 41.6 | 24.2 |
| | SnapKV[38] | **87.8** | 87.6 | 83.2 | 88.0 | 56.2 | 81.8 | **81.2** | 78.0 | 68.4 | 51.6 | 33.0 | 19.4 |
| | KeyDiff[52] | **87.8** | 87.2 | 84.2 | 75.0 | 45.0 | 21.8 | **81.2** | 80.6 | 76.6 | 64.8 | 45.4 | 24.6 |

Table 18: **Question Answering 2**

| Model | Method | Ruler 4k | | | | | | Ruler 16k | | | | | |
|---|---|---|---|---|---|---|---|---|---|---|---|---|---|
| | | 0% | 10% | 25% | 50% | 75% | 90% | 0% | 10% | 25% | 50% | 75% | 90% |
| Qwen3-8B | EA (Ours) | **63.4** | 62.6 | 62.4 | 58.8 | 50.6 | 35.6 | **58.8** | 58.4 | 58.2 | 55.6 | 48.8 | 35.4 |
| | TOVA[51] | **63.4** | 63.0 | 60.0 | 54.6 | 40.2 | 26.6 | **58.8** | 57.2 | 55.4 | 49.4 | 38.6 | 27.0 |
| | SnapKV[38] | **63.4** | 61.6 | 58.2 | 51.4 | 41.8 | 27.4 | **58.8** | 55.6 | 52.0 | 45.4 | 33.2 | 26.6 |
| | KeyDiff[52] | **63.4** | 56.4 | 46.0 | 27.8 | 13.4 | 10.6 | **58.8** | 51.0 | 43.0 | 31.0 | 18.6 | 12.8 |
| Gemma3-12B | EA (Ours) | **61.0** | 60.0 | 59.4 | 55.2 | 42.8 | 33.8 | **54.8** | 52.0 | 52.2 | 47.0 | 33.6 | 26.4 |
| | TOVA[51] | **61.0** | 60.8 | 59.0 | 54.0 | 45.6 | 35.2 | **54.8** | 54.6 | 52.8 | 47.0 | 37.6 | 30.4 |
| | SnapKV[38] | **61.0** | 59.4 | 55.4 | 52.4 | 43.6 | 31.0 | **54.8** | 54.6 | 49.8 | 42.8 | 32.6 | 24.8 |
| | KeyDiff[52] | **61.0** | 59.0 | 51.8 | 39.0 | 21.4 | 14.4 | **54.8** | 53.2 | 47.2 | 33.0 | 19.2 | 12.2 |
| Llama3.1-8B | EA (Ours) | **62.8** | 62.6 | 61.2 | 58.6 | 50.2 | 42.2 | **57.0** | 57.8 | 55.2 | 54.4 | 43.0 | 29.4 |
| | TOVA[51] | **62.8** | 61.4 | 59.8 | 54.2 | 42.8 | 28.2 | **57.0** | 55.6 | 54.8 | 48.6 | 36.8 | 28.4 |
| | SnapKV[38] | **62.8** | 62.0 | 58.6 | 62.0 | 37.6 | 59.6 | **57.0** | 54.0 | 54.4 | 45.2 | 34.4 | 28.8 |
| | KeyDiff[52] | **62.8** | 63.4 | 61.0 | 49.4 | 31.6 | 18.2 | **57.0** | 56.8 | 55.2 | 52.8 | 45.6 | 34.2 |

Table 19: **Variable Tracking**

| Model | Method | Ruler 4k | | | | | | Ruler 16k | | | | | |
|---|---|---|---|---|---|---|---|---|---|---|---|---|---|
| | | 0% | 10% | 25% | 50% | 75% | 90% | 0% | 10% | 25% | 50% | 75% | 90% |
| Qwen3-8B | EA (Ours) | **100.0** | 100.0 | 99.9 | 87.3 | 14.0 | 0.7 | **100.0** | 100.0 | 100.0 | 100.0 | 100.0 | 99.9 |
| | TOVA[51] | **100.0** | 100.0 | 100.0 | 100.0 | 96.3 | 15.4 | **100.0** | 100.0 | 100.0 | 100.0 | 100.0 | 88.5 |
| | SnapKV[38] | **100.0** | 98.8 | 94.2 | 81.2 | 33.2 | 8.6 | **100.0** | 99.9 | 99.6 | 99.0 | 94.7 | 66.4 |
| | KeyDiff[52] | **100.0** | 100.0 | 100.0 | 100.0 | 98.9 | 32.9 | **100.0** | 100.0 | 100.0 | 100.0 | 100.0 | 79.9 |
| Gemma3-12B | EA (Ours) | **99.7** | 99.7 | 99.6 | 99.4 | 98.3 | 93.8 | **96.4** | 94.0 | 93.8 | 90.9 | 84.0 | 80.3 |
| | TOVA[51] | **99.7** | 99.6 | 99.7 | 99.6 | 93.4 | 27.7 | **96.4** | 96.4 | 96.6 | 96.2 | 94.3 | 84.9 |
| | SnapKV[38] | **99.7** | 99.0 | 97.8 | 89.2 | 70.2 | 46.2 | **96.4** | 93.5 | 91.6 | 90.8 | 87.8 | 74.1 |
| | KeyDiff[52] | **99.7** | 99.6 | 99.6 | 99.6 | 97.6 | 80.0 | **96.4** | 96.6 | 96.2 | 95.3 | 93.3 | 82.6 |
| Llama3.1-8B | EA (Ours) | **99.9** | 99.8 | 99.8 | 88.2 | 66.6 | 35.6 | **99.8** | 99.7 | 99.5 | 99.5 | 96.2 | 76.3 |
| | TOVA[51] | **99.9** | 99.9 | 99.9 | 99.9 | 96.6 | 20.4 | **99.8** | 99.8 | 99.8 | 99.8 | 99.8 | 94.1 |
| | SnapKV[38] | **99.9** | 99.6 | 93.4 | 97.7 | 59.8 | 53.7 | **99.8** | 96.2 | 95.2 | 93.4 | 81.4 | 53.4 |
| | KeyDiff[52] | **99.9** | 99.9 | 99.9 | 99.9 | 99.6 | 77.6 | **99.8** | 99.8 | 99.7 | 99.6 | 98.7 | 94.6 |

Table 20: **Average**

| Model | Method | Ruler 4k | | | | | | Ruler 16k | | | | | |
|---|---|---|---|---|---|---|---|---|---|---|---|---|---|
| | | 0% | 10% | 25% | 50% | 75% | 90% | 0% | 10% | 25% | 50% | 75% | 90% |
| Qwen3-8B | EA (Ours) | **95.3** | 88.5 | 77.2 | 52.2 | 22.5 | 11.3 | **93.0** | 93.1 | 93.2 | 92.7 | 85.6 | 62.7 |
| | TOVA[51] | **95.3** | 89.0 | 82.5 | 77.6 | 62.4 | 24.7 | **93.0** | 88.3 | 81.7 | 76.2 | 68.7 | 52.4 |
| | SnapKV[38] | **95.3** | 92.6 | 84.0 | 55.7 | 33.1 | 19.2 | **93.0** | 90.1 | 81.5 | 62.8 | 41.7 | 26.8 |
| | KeyDiff[52] | **95.3** | 93.8 | 89.4 | 78.6 | 64.4 | 37.9 | **93.0** | 88.9 | 82.9 | 74.5 | 66.9 | 53.1 |
| Gemma3-12B | EA (Ours) | **95.2** | 95.2 | 94.9 | 92.7 | 78.2 | 53.6 | **86.0** | 82.8 | 81.7 | 76.6 | 60.5 | 41.8 |
| | TOVA[51] | **95.2** | 89.7 | 81.1 | 76.5 | 58.1 | 25.3 | **86.0** | 79.7 | 72.6 | 62.5 | 46.8 | 32.7 |
| | SnapKV[38] | **95.2** | 82.9 | 72.0 | 54.8 | 40.3 | 30.1 | **86.0** | 74.1 | 62.8 | 46.4 | 37.3 | 31.4 |
| | KeyDiff[52] | **95.2** | 94.3 | 90.6 | 79.8 | 62.0 | 34.3 | **86.0** | 81.8 | 78.6 | 72.6 | 58.6 | 37.2 |
| Llama3.1-8B | EA (Ours) | **95.7** | 92.7 | 95.3 | 76.6 | 55.7 | 29.0 | **93.4** | 93.4 | 92.8 | 86.0 | 66.4 | 25.5 |
| | TOVA[51] | **95.7** | 93.2 | 87.3 | 76.2 | 63.3 | 37.5 | **93.4** | 90.9 | 86.1 | 77.9 | 68.4 | 59.2 |
| | SnapKV[38] | **95.7** | 95.5 | 81.8 | 88.8 | 43.4 | 63.2 | **93.4** | 89.4 | 82.0 | 68.0 | 43.1 | 25.6 |
| | KeyDiff[52] | **95.7** | 94.7 | 91.6 | 85.5 | 72.9 | 61.1 | **93.4** | 92.1 | 88.4 | 82.6 | 74.9 | 66.5 |

