# OpenReview forum: "Expected Attention: KV Cache Compression by Estimating Attention From Future Queries Distribution"
_ICLR.cc/2026/Conference — Submitted to ICLR 2026_

### Official Review · Reviewer_Vi4u · 2025-10-22

**Soundness:** 3
**Presentation:** 3
**Contribution:** 3
**Rating:** 6
**Confidence:** 4

**Summary:**

This paper introduces Expected Attention, a training-free KV cache compression method that leverages the Gaussian properties of LLM activations to estimate the importance of KV pairs for future tokens. The method is evaluated across multiple benchmarks and model families and shows competitive or superior performance compared to existing baselines. While the idea is novel and well-motivated, the paper has several weaknesses that need to be addressed before acceptance.

**Strengths:**

1.The idea of using Expected Attention to estimate future KV importance without requiring future queries is innovative and theoretically grounded.
2.Extensive experiments across prefilling and decoding phases, multiple models, and benchmarks (LongBench, Ruler, NIAH, Aime25, MATH-500) demonstrate the method's robustness.
3.The authors commit to releasing code and provide detailed experimental setups.

**Weaknesses:**

1.The Gaussian assumption of hidden states and queries is central to the method but is not rigorously validated across all layers, models, or tasks. The paper should include more quantitative evidence (e.g., normality tests) beyond visual fits in the appendix.
2.The approximation of the RoPE matrix over a fixed future window T=512 is arbitrary and not well-motivated. How sensitive is the method to the choice of T? An ablation study is necessary.
3.The performance on Gemma3-12B in Table 3 is inconsistent and sometimes worse than TOVA . This should be discussed and analyzed.
4.The computational overhead of computing query statistics and expected attention scores is not quantified. How does this impact latency, especially during decoding?
5.The contribution of each component (e.g., Gaussian assumption, RoPE averaging, adaptive per-head compression) is not isolated. Ablation studies are needed to justify the design choices.

**Questions:**

See Weakness

---

> ### Author Response · Authors · 2025-11-21
> **Reply (1/2)**
>
> Thank you for your detailed and insightful review. We are glad that you found our core idea of Expected Attention "innovative" and supported by extensive experimental evidence. We fully agree that addressing the raised weaknesses is crucial for acceptance, and we are committed to performing the requested analysis to strengthen our submission.
>
> ### Weaknesses
>
> **1. Clarification about the Gaussian assumption of hidden states**
>
> Thank you, we agree this point deserved a clearer explanation.
> - We clarify that we only assume **approximate Gaussianity** (lines #160-165): hidden states and queries typically exhibit unimodal and roughly symmetric distributions, rather than strict normality. Crucially, **Expected Attention does not rely on exact Gaussianity**. The method only requires unimodality to make the expected-value tractable.
>
> - Additionally, we note that this assumption is **aligned with prior accepted work** such as Teal et al. [6], which adopts the same modeling choice.
>
> We are sorry about the confusion and have revised the paper in Section 2.2 and Appendix B to make this more explicit.
>
> **2. The approximation of the RoPE matrix over a fixed future window T=512 is arbitrary and not well-motivated. How sensitive is the method to the choice of T? An ablation study is necessary.**
>
> Thank you for pointing this out, we agree that this deserves more validation. Following your suggestion, we have conducted an **ablation to assess sensitivity to T**, that we report here and in a new Section 5.1. The minimal performance drop observed across different models when reducing $T$ from $1024$ to $512$ or even $256$ is justifying the practical choice of $T=512$ that we used in our experiments.
>
> | Model | T=1024 | T=512 | T=256 | T=128 |
> | :--- | :--- | :--- | :--- | :--- |
> | **Llama3.1 8B** | 92.1 | 92.2 | 91.9 | 91.8 |
> | **Qwen3 8B** | 94.8 | 94.7 | 94.7 | 94.8 |
> | **Gemma3 12 B** | 92.7 | 92.7 | 92.7 | 92.7 |
>
> **3. The performance on Gemma3-12B in Table 3 is inconsistent and sometimes worse than TOVA. This should be discussed and analyzed.**
>
> Thank you for highlighting this. We investigated this point in more depth and found that this anomalous behavior is **isolated to the Gemma-3-12B model on this specific benchmark**. Importantly, this issue does not appear in any of the other benchmarks or model families we evaluated.
> These results for **Gemma3-12B on LongBench** exhibit behaviors that differ from other models. Specifically, **all compression methods show an initial increase in average score** at the $10\%$ and $25\%$ compression ratios compared to the $0\%$ baseline. This unexpected gain suggests that removing a small fraction of the least-important Key-Value pairs effectively prunes noisy or redundant information for this model, thereby enhancing performance.
>
> While the effect does not alter the paper's overall conclusions, we agree it's worth noting. In the revision, we have added a clarification stating that **the anomaly is isolated to this particular setting**, it does not occur in any other tasks or models, and its underlying cause remains an open question that we plan to investigate further. We believe this is the most transparent way to present the result without over-interpreting it.
>
>
> **4. The computational overhead of computing query statistics and expected attention scores is not quantified.**
>
> While EA introduces only **two matrix multiplications per layer** during scoring and an initial computation of the statistics, we conducted two additional analyses:
>
> 1. We provide a **FLOPs analysis** performed following **[1]**. We compute the model FLOPs for Llama3.1 and the FLOPs resulting from Expected Attention overhead. Results show that this accounts for just **0.5% of the model FLOPs**, a **negligible computational overhead.
>
> 2. To support this with empirical data, we include **new latency measurements** obtained using our released PyTorch implementation at a 50% compression ratio with Llama3.1-8B:
>    - **Prefill latency increases by only ~2%**
>    - **Generation latency decreases by ~25%** due to smaller KV Cache.
>
> We believe these results are enough to validate the computational efficiency of Expected Attention, confirming its negligible theoretical overhead** and demonstrating a significant empirical speedup in the critical generation phase.
>
> These numbers serve as an upper bound since highly optimized kernels (e.g., FlashAttention-style fused ops) would reduce overhead further. We added these results in the main paper as we believe they are a valid addition to the method.
>
> Finally, we would like to point out that together with the paper, we are releasing a **comprehensive benchmarking library** that reproduces all scores, also for the baselines. This ensures transparency and lets reviewers inspect the exact implementation. We hope this library can serve as a standard reference for KV Cache compression research in the future.

---

> > ### Author Response · Authors · 2025-11-21
> > **Reply (2/2)**
> >
> > **5. The contribution of each component (e.g., Gaussian assumption, RoPE averaging, adaptive per-head compression) is not isolated. Ablation studies are needed to justify the design choices.**
> >
> > We agree that these components deserve better investigation, and we performed additional ablations for the key design choices after your suggestion:
> >
> > - **Gaussian assumption:** Discussed in (1) above.
> >
> > - **RoPE averaging / T value:** Discussed in (2). We performed additional experiments and added this to the main manuscript.
> >
> > - **Adaptive compression:** We ran additional experiments and tested a **uniform compression baseline** (no per-head adaptation) on Ruler 4K. Results confirms that adaptive compression significantly contributes to performance.
> >
> > - In addition to this, we performed another ablation on the covariance term, showing it might be possibly be removed in the future and simplify the method even further.
> >
> > | Model | EA | w/o Adaptive | w/o Covariance |
> > | :--- | :--- | :--- | :--- |
> > | **Llama3.1 8B** | 92.2 | 86.5 | 90.6 |
> > | **Qwen3 8B** | 94.7 | 86.6 | 94.7 |
> > | **Gemma3 12 B** | 92.7 | 88.2 | 92.6 |
> >
> >
> > Thank you for actually taking time to read and review the paper in detail. We believe the new ablation studies that we added following your comments improved the quality of our paper significantly. We remain available for further clarification or discussion if needed.
> >
> > ### References
> > **[1]** Jordan Hoffmann et al., **"Training Compute-Optimal Large Language Models"** in Advances in Neural Information Processing Systems (2022).
> >
> > **[6]** James Liu et al., **"Training-Free Activation Sparsity in Large Language Models"** in The Thirteenth International Conference on Learning Representations (2025).

---

> > > ### Comment · Reviewer_Vi4u · 2025-11-26
> > >
> > > I have carefully read the rebuttal and acknowledge the authors' efforts in their rebuttal. However, I believe the rebuttal largely confirms my initial assessment, and thus I do not find sufficient reason to adjust my original score.

---

> ### Comment · Area_Chair_7B1b · 2025-11-23
>
> Dear reviewer,
>
> Thanks for your time and effort in reviewing ICLR2026 submissions. The authors have submitted their responses to your review. Please take the time to read and raise your further comments, and discuss with the authors.
>
> Best regards,
>
> AC

---

### Official Review · Reviewer_6DBQ · 2025-10-23

**Soundness:** 3
**Presentation:** 3
**Contribution:** 3
**Rating:** 4
**Confidence:** 3

**Summary:**

The authors propose a form of KV cache compression which operates based on measuring the residual connection contribution of a KV pair instead of the attention score. They predict attention contribution implicitly by modeling the distribution of future attention values.

**Strengths:**

- The approach is novel and interesting since most prior works focus on attention scores.

- The approach intuitively makes sense, because the contribution is from both the attention scores and the value vector, while prior work only focuses on the attention scores. If the value vector norm is 0 or close to zero, these high attention scores are null.

**Weaknesses:**

- Why was RULER only considered up to 16K. Most of the models under consideration in table 1 can go well past 16 natively.

---

Overall, I would be interested to see how the performance holds up over the baseline models for the longer RULER context lengths and on the full subset of RULER tasks. For instance, the harder retrieval tasks like multi-key 3 seem to be omitted from the current set of experiments. If the authors could show that the method holds up for these harder settings, it would be very compelling.

**Questions:**

- What is meant by "activation value" in figure 1? Is it the dot product value before the exponential? Or something else

- For clarity, can you add a derivation of how you get the expression on teh RHS of equation 7?

- Why do SnapKV and TOVA show odd patterns of missing the needle at around 100-115K context lengths in Figure 3?

---

> ### Author Response · Authors · 2025-11-21
> **Reply (1/2)**
>
> Thank you for your thoughtful review and for recognizing the novelty and intuitive strengths of our approach, particularly focusing on the residual connection contribution of a KV pair. We really appreciated your positive assessment of the novelty and contribution of the method.
>
> We have updated our manuscript and appendix with additional experiments based on your feedback, and we attempt here to provide answers to all your doubts. Changes to manuscript are highlighted in orange.
>
> **Why was RULER only considered up to 16K. Most of the models under consideration in table 1 can go well past 16 natively.**
>
> We agree that evaluating performance on longer RULER contexts is valuable. In our work, we chose the 4K and 16K setting because it captures the regime where KV-cache eviction strategies begin to matter, while keeping the evaluation computationally feasible given the large number of models and methods compared across all benchmarks. Importantly, the broader long-context generalization of Expected Attention is already demonstrated in the paper through other benchmarks such as LongBench and Needle-in-a-Haystack which **include contexts up to 125K**. We think these tasks, together with the decoding evaluations, collectively show that Expected Attention maintains strong performance in substantially longer-context scenarios.
>
> **For instance, the harder retrieval tasks like Ruler multi-key 3 seem to be omitted from the current set of experiments. If the authors could show that the method holds up for these harder settings, it would be very compelling.**
>
> Following your comment, we have conducted and now provided the full breakdown of RULER sub-tasks, including the more challenging retrieval tasks like Multi-key 3. We show that **Multikey3 is one of the tasks where EA outperforms other baselines with a larger gap**. We report the full breakdown in Appendix D, and we show here the results on the Multi-key 3 subset.
>
>
>
> | Model | Method | **Ruler 4k** | | | | | | **Ruler 16k** | | | | | |
> | :---: | :---: | :---: | :---: | :---: | :---: | :---: | :---: | :---: | :---: | :---: | :---: | :---: | :---: |
> | | | **0%** | **10%** | **25%** | **50%** | **75%** | **90%** | **0%** | **10%** | **25%** | **50%** | **75%** | **90%** |
> | :---: | :---: | :---: | :---: | :---: | :---: | :---: | :---: | :---: | :---: | :---: | :---: | :---: | :---: |
> | ***Qwen*** | EA (Ours) | **100.0** | 87.2 | 47.2 | 9.8 | 0.2 | 0.0 | **99.6** | 99.8 | 99.8 | 99.0 | 45.4 | 0.0 |
> |  | TOVA | **100.0** | 50.4 | 12.2 | 0.4 | 0.0 | 0.0 | **99.6** | 63.8 | 24.4 | 4.0 | 0.6 | 0.0 |
> |  | SnapKV | **100.0** | 87.8 | 58.4 | 20.0 | 1.6 | 0.0 | **99.6** | 88.8 | 59.6 | 18.0 | 3.8 | 1.0 |
> |  | KeyDiff | **100.0** | 92.8 | 67.8 | 18.6 | 2.0 | 0.0 | **99.6** | 82.8 | 50.2 | 14.2 | 0.6 | 0.0 |
> | | | | | | | | | | | | | | |
> | ***Gemma*** | EA (Ours) | **99.8** | 99.6 | 99.2 | 86.8 | 11.8 | 0.0 | **61.6** | 45.8 | 41.4 | 25.0 | 8.8 | 0.0 |
> |  | TOVA | **99.8** | 65.8 | 8.8 | 0.0 | 0.0 | 0.0 | **61.6** | 26.8 | 8.8 | 0.6 | 0.0 | 0.0 |
> |  | SnapKV | **99.8** | 93.2 | 68.2 | 28.8 | 2.4 | 0.0 | **61.6** | 56.2 | 41.2 | 9.6 | 2.0 | 0.6 |
> |  | KeyDiff | **99.8** | 92.0 | 68.2 | 7.2 | 0.0 | 0.0 | **61.6** | 32.4 | 21.4 | 10.6 | 0.0 | 0.0 |
> | | | | | | | | | | | | | | |
> | ***Llama*** | EA (Ours) | **99.8** | 100.0 | 99.8 | 27.0 | 0.2 | 0.0 | **99.2** | 99.2 | 99.0 | 54.6 | 10.8 | 0.0 |
> |  | TOVA | **99.8** | 74.6 | 33.4 | 2.6 | 0.0 | 0.0 | **99.2** | 77.2 | 39.2 | 8.2 | 1.0 | 0.4 |
> |  | SnapKV | **99.8** | 99.8 | 55.2 | 84.0 | 1.6 | 0.0 | **99.2** | 86.6 | 60.2 | 18.2 | 3.6 | 1.0 |
> |  | KeyDiff | **99.8** | 87.2 | 53.2 | 11.0 | 0.0 | 0.0 | **99.2** | 82.8 | 43.2 | 6.8 | 0.0 | 0.0 |

---

> > ### Author Response · Authors · 2025-11-21
> > **Reply (2/2)**
> >
> > ### Questions
> >
> > **What is meant by "activation value" in Figure 1? Is it the dot product value before the exponential? Or something else?**
> >
> > The "Activation Value" in Figure 1 refers to the pre-activation input to the transformation function. Specifically, for the hidden states plots, it refers to the scalar values of each dimension in the hidden state vectors $h \in \mathbb{R}^h$. For the query plots, it refers to the scalar values of the projected query vectors $q \in \mathbb{R}^d$. It is the value before the dot product for attention computation. Thanks for asking this, we added a more detailed description in the manuscript.
> >
> > **For clarity, can you add a derivation of how you get the expression on the RHS of equation 7?**
> >
> > We hope to clarify this in the following and in the new Appendix F. Equation (7) follows directly from the [moment-generating function of a Gaussian](https://statproofbook.github.io/P/norm-mgf.html), applied to the random variable $q^\top k / \sqrt{d}$. We add a short proof here:
> >
> > * The term in the exponent,
> >   $X = \frac{\mathbf{q}^\top \mathbf{k}_i}{\sqrt{d}}$
> >   is a Gaussian random variable, $X \sim \mathcal{N}(\mu_X, \sigma_X^2)$
> >
> > * The mean of X is
> >   $\mu_X = \mathbb{E}[X] = \frac{\bar{\mu}_q^\top \mathbf{k}_i}{\sqrt{d}}$
> >
> > * The variance of X is
> >   $ \sigma_X^2 = \mathrm{Var}(X) = \frac{\mathbf{k}_i^\top \bar{\Sigma}_q \mathbf{k}_i}{d}$
> >
> > The expectation $\mathbb{E}[\exp(X)]$ is computed using the Moment-Generating Function (MGF)
> > $M_X(t)$ = $\mathbb{E}[e^{tX}]$ of a Gaussian distribution, evaluated at (t = 1).
> > Since
> > $M_X(t)$ = $\exp\left(\mu_X t + \frac{1}{2}\sigma_X^2 t^2\right)$,
> > setting (t=1) yields:
> >
> > $
> > \mathbb{E}[\exp(X)] = \exp\left(\mu_X + \frac{\sigma_X^2}{2}\right).
> > $
> >
> > Substituting $\mu_X$ and $\sigma_X^2$ recovers Equation 7.
> >
> > **Why do SnapKV and TOVA show odd patterns of missing the needle at around 100-115K context lengths in Figure 3?**
> >
> > We appreciate this observation. After further investigation, we are not certain about the cause of this behavior. Our best hypothesis is that these methods may exhibit heuristic-level instability at very long context lengths, but we do not have definitive evidence. We have therefore chosen not to over-interpret these patterns and simply report them faithfully.
> >
> > Thank you for actually taking time to read and review the paper in detail. We hope our answers clarified your doubts, as we think the detailed evaluation results we provided following your concerns improved the readability and quality of our paper. We remain available for further clarification or discussion if needed.

---

> > > ### Comment · Reviewer_6DBQ · 2025-11-27
> > >
> > > Thank you for responding to my questions in detail. I think this method is completely novel and interesting. It seems useful for the case where the KV importance is not directly indicated by the trailing queries in the prompt (like SnapKV utilizes). I will raise my score.

---

> ### Comment · Area_Chair_7B1b · 2025-11-23
>
> Dear reviewer,
>
> Thanks for your time and effort in reviewing ICLR2026 submissions. The authors have submitted their responses to your review. Please take the time to read and raise your further comments, and discuss with the authors.
>
> Best regards,
>
> AC

---

### Official Review · Reviewer_myPE · 2025-10-31

**Soundness:** 2
**Presentation:** 3
**Contribution:** 3
**Rating:** 4
**Confidence:** 4

**Summary:**

The author proposed a KV eviction method based on query vector statistics. It captures outstanding tokens in the KV cache in a query-statistics-aware manner rather than simply using a pooled query or a particular query.

**Strengths:**

The method implementation is pretty simple.

**Weaknesses:**

- I am not sure this method is really helpful
- The mathematical analysis is not sufficiently intuitive and expressive to convince the reader to use covariant statistics of query vectors.
- More importantly, the empirical results are not significantly improved compared to baselines in LongBench.
- Even in the Llama 3.1, the downstream task performance is drastically dropped compared to baselines (25.5 vs. 66.5), which is a clear signal that something is wrong with the method (or experiment).

**Questions:**

- What is the latency overhead of computing statistics of queries?

---

> ### Author Response · Authors · 2025-11-21
> **Reply (1/2)**
>
> Thank you for taking time to review our paper and for the positive assessment of our method's simplicity. We address the concerns you raised about covariance and empirical results in the following answers and in changes to the manuscript, highlighted in orange.
>
>
> **1. Concern about whether the method is helpful**
>
> We appreciate this concern and would like to clarify the contribution of Expected Attention. We believe it provides substantial value to KV cache compression, **a critical research area** and challenge for efficient LLMs. Our core contribution is a principled, training-free mechanism that identifies which Key-Value pairs matter most for future computation, something that has lacked principled treatment in prior work.
>
> The method's usefulness is reflected in the positive feedback from other reviewers, who highlighted its "**novelty**," "**innovation**," and "**theoretical grounding**" (R5kT, 6DBQ, Vi4u). More importantly, our **extensive empirical validation across multiple models and benchmarks** demonstrates that Expected Attention **consistently achieves superior accuracy-compression trade-offs** compared to existing training-free baselines, the defining success metric for this task. Beyond accuracy, our efficiency new analysis shows that the method introduces **negligible computational overhead** while delivering  generation speedups through reduced memory footprint.
>
> **2. Even in the Llama 3.1, the downstream task performance is drastically dropped compared to baselines (25.5 vs. 66.5), which is a clear signal that something is wrong with the method (or experiment).**
>
> We appreciate this concern and provide two clarifications.
>
> - The reported drop occurs at **extreme compression (90%)**, where **all methods degrade severely**. At this ratio, baseline performance is already very low, and this regime is **not the intended operating point** for KV compression methods [4]. The meaningful comparison is at **moderate compression** or better at the compression regime where the downstream performance does not drop significantly (some works quantified this with 90% of the initial performance [4]). In this regime, **EA outperforms alternatives**. We report a 90% compression ratio for completeness but it is not the target of the method.
>
> - The highlighted drop concerns KeyDiff, which is particularly suited to synthetic, high-redundancy tasks such as RULER, where its score-based mechanism aligns with strong intra-context similarity [5]. As shown in the paper, on **broader and more diverse datasets (e.g., LongBench), EA significantly outperforms KVZip** at standard compression ratios. However, we agree this should be clarified and added this observation in the experimental results section in 4.1.
>
> Finally but very importantly, we would like to point out that together with the paper, we released a **comprehensive benchmarking library** that reproduces **all scores, also for the baselines**. This ensures transparency and lets reviewers inspect the exact implementation. We hope this library can serve as a standard reference for KV Cache compression research in the future.
>
> **3. Improvement of scores on Longbench.**
>
> We agree we should make this more clear. We clarify that **tradeoff, not absolute score, is the key metric** for KV Cache compression. The goal of KV compression is to **preserve accuracy under a given compression ratio**, and across LongBench and RULER our method consistently achieves the best accuracy–compression tradeoff, particularly in the practical, low degradation regime, which is the one where performance does not degrade significantly with respect to the uncompressed baseline [4].
>
> **4. The mathematical analysis concerning the use of covariance.**
>
> Thank you for pointing this out. We agree and revised Section 2.2 to strengthen the intuition:
>
> - **Mean term ($\mu_q$):** captures relevance to the **average future query**.
>
> - **Covariance term ($\Sigma_q$):** captures **uncertainty and diversity in future queries**. **Keys that align with directions of high query variability receive higher scores**, making the selection robust to a range of possible upcoming queries, not just the mean.
>
> Additionally, following your suggestion we performed an **additional ablation study** investigating the impact of the covariance term on the method, that we report here and in a new Section 5.1. While its removal causes a noticeable performance drop (92.2 → 90.6) for Llama, the effect is minimal for Qwen and Gemma. We conjecture this reduced dependency is due to their QK normalization. This finding is particularly encouraging as it suggests that for models employing QK normalization, we could **safely omit the covariance term in future implementations**, thereby making the method even simpler.
>
> | Model | Eexpected Attention (ours)  | w/o Covariance |
> | :--- | :--- | :--- |
> | **Llama3.1 8B** | 92.2  | 90.6 |
> | **Qwen3 8B** | 94.7 | 94.7 |
> | **Gemma3 12 B** | 92.7 | 92.6 |

---

> ### Author Response · Authors · 2025-11-21
> **Reply (2/2)**
>
> ### Questions
>
> **What is the latency overhead of computing statistics of queries?**
>
> While EA introduces only **two matrix multiplications per layer** during scoring and an initial computation of the statistics, we conducted two additional analyses following your advice:
>
> 1. We provide a **FLOPs analysis** performed following [1]. We compute the model FLOPs for Llama3.1 and the FLOPs resulting from Expected Attention overhead. Results show that this accounts for just **0.5% of the model FLOPs**, a **negligible computational overhead**.
>
> 2. To support this with empirical data, we include **new latency measurements** obtained using our released PyTorch implementation at a 50% compression ratio with Llama3.1-8B:
>    - **Prefill latency increases by only ~2%**
>    - **Generation latency decreases by ~25%** due to smaller KV Cache.
>
> | Phase | No Compression | EA (50% comp.) | Variation (abs) | Variation (%) |
> | :--- | :---: | :---: | :---: | :---: |
> | Prefilling | 15.25 $\pm$ 0.02 | 15.52 $\pm$ 0.02 | +0.27 | +1.74 |
> | Generation | 4.33 $\pm$ 0.00 | 3.22 $\pm$ 0.04 | -1.11 | -25.58 |
> | **Total** | 19.58 $\pm$ 0.03 | 18.74 $\pm$ 0.03 | -0.84 | **-4.30** |
>
> We believe these results are enough to **validate the computational efficiency of Expected Attention**, confirming its negligible theoretical overhead and demonstrating a significant empirical speedup in the critical generation phase.
>
> These numbers serve as an upper bound since highly optimized kernels (e.g., FlashAttention-style fused ops) would reduce overhead further. We added these results in the main paper in Section 4.4 as we believe they are a valid addition to the method.
>
> Thank you for your comments, we believe the additions that we made to the paper following your concerns improved the quality significantly. We hope we clarified all your doubts and remain available for further clarification or discussion if needed.
>
> ### References
> **[1]** Jordan Hoffmann et al., **"Training Compute-Optimal Large Language Models"** in Advances in Neural Information Processing Systems (2022).
>
> **[4]** Bhaskar et al.,  **"How Many KVs Do You Need for Effective Long-Context LMs?"**, 2025
>
> **[5]** Park et. al, **"KeyDiff: Key Similarity-Based KV Cache Eviction for Long-Context LLM Inference in Resource-Constrained Environments"**, 2025
>
> ---

---

> ### Comment · Area_Chair_7B1b · 2025-11-23
>
> Dear reviewer,
>
> Thanks for your time and effort in reviewing ICLR2026 submissions. The authors have submitted their responses to your review. Please take the time to read and raise your further comments, and discuss with the authors.
>
> Best regards,
>
> AC

---

### Official Review · Reviewer_R5kT · 2025-11-01

**Soundness:** 3
**Presentation:** 3
**Contribution:** 3
**Rating:** 6
**Confidence:** 3

**Summary:**

This paper presents Expected Attention, a training-free approach for compressing the KV cache during large language model inference. It estimates each key’s expected contribution to future attention by assuming hidden states follow a Gaussian distribution and analytically computing expected attention scores using moment-generating functions. By ranking and pruning low-impact KV pairs based on these scores, the method reduces memory use while maintaining accuracy. Experiments on long-context and reasoning benchmarks (LongBench, Ruler, MATH-500, AIME25) show that Expected Attention matches or surpasses prior training-free baselines such as SnapKV and TOVA, achieving up to 50% compression with minimal performance degradation. Limitations include manual compression ratio tuning and lack of kernel-level optimization.

**Strengths:**

### Novel yet practical idea:
The paper introduces a theoretically grounded, training-free method for KV cache compression based on the expected attention formulation, which is both conceptually elegant and computationally feasible.

### Solid theoretical foundation:
The derivation using Gaussian assumptions and moment-generating functions is mathematically sound and provides clear intuition on how expected attention can approximate future query behavior.

### Extensive empirical validation across model families:
The method is tested on a wide range of LLMs, including LLaMA-3.1-8B, Qwen3-8B, Gemma-3-12B, and reasoning-focused models such as Qwen-R1-1.5B/3B and Nemotron-14B, demonstrating robustness across architectures and scales.

### Comprehensive benchmark coverage:
Evaluations span both long-context tasks (LongBench, Ruler, Needle-in-a-Haystack) and reasoning benchmarks (MATH-500, AIME25), showing that Expected Attention maintains accuracy even under substantial compression ratios.

**Weaknesses:**

### Lack of Efficiency Comparison and Runtime Analysis:
While the paper presents an insightful theoretical formulation and strong accuracy results, its evaluation of practical efficiency is limited. Although Figure 5 reports the peak memory usage of Expected Attention under varying sequence lengths and compression ratios, there is no direct comparison of memory footprint against prior baselines (e.g., SnapKV, TOVA, KeyDiff, KNorm, StreamingLLM). Moreover, inference latency or throughput measurements are not provided, making it difficult to assess the real-world efficiency gains. Given that the primary motivation of KV cache compression is to reduce memory and runtime overhead, a more thorough empirical analysis of these aspects would significantly strengthen the paper.

### Minor point:
1. Missing caption for Figure 5:
The paper includes two subfigures, (a) and (b), showing peak memory usage under different conditions, but the main caption “Figure 5:” itself is missing. Each figure in the paper should include a clear and complete caption describing its content, in accordance with the conference formatting guidelines.

2. Incorrect caption placement for Figure 4:
The caption for Figure 4 appears above the figure, likely to align with the neighboring Table 2 caption, since table captions are required to appear above tables. However, according to the official ICLR formatting rule (`The figure number and caption always appear after the figure`), figure captions must be placed below the figure. While the intent to maintain visual alignment is understandable, adhering to the guideline would ensure consistency and compliance with the submission standards.

**Questions:**

### Batch inference comparison:
All reported experiments appear to use batch size 1, focusing on single-sequence inference. In real-world deployment scenarios, LLM inference often runs with larger batch sizes for better throughput and GPU utilization. How would Expected Attention perform when scaling batch size — both in terms of GPU memory consumption and inference latency — compared to prior training-free baselines? Evaluating this aspect would provide stronger evidence of the method’s practicality for real-world applications.

### Computation efficiency comparison:
How does Expected Attention compare to other training-free KV cache compression methods in terms of actual GPU memory consumption, inference latency, and throughput? Including such comparisons would clarify the real-world efficiency benefits of the method.

---

> ### Author Response · Authors · 2025-11-21
> **Reply (1/2)**
>
> Thank you for taking time to review the paper and for the constructive and positive assessment of our work. We are glad that you found Expected Attention “novel”,  “theoretically grounded”, and “extensively validated”. The main points of concern relate to practical efficiency, which we address below with clarifications and changes to manuscript, highlighted in orange.
>
> **1. While the paper presents an insightful theoretical formulation and strong accuracy results, its evaluation of practical efficiency is limited. Although Figure 5 reports the peak memory usage of Expected Attention under varying sequence lengths and compression ratios, there is no direct comparison of memory footprint against prior baselines (e.g., SnapKV, TOVA, KeyDiff, KNorm, StreamingLLM).**
>
> We agree that we should make this point more clear. All KV-compression methods have a **memory footprint determined by the compression ratio**: if two methods retain the same proportion of tokens, their KV footprint is almost identical. Our contribution focuses on which tokens to retain. Under equal compression ratios, EA consistently yields higher accuracy than prior training-free baselines, meaning it delivers **better accuracy–memory trade-offs at matched memory budgets**. Therefore, we did not show the memory footprint of other methods as they would all have the same for each compression ratio. **The superior accuracy of EA for any given memory budget is the core efficiency benefit.**
>
> **2.  Inference latency or throughput measurements are not provided, making it difficult to assess the real-world efficiency gains. Given that the primary motivation of KV cache compression is to reduce memory and runtime overhead, a more thorough empirical analysis of these aspects would significantly strengthen the paper.**
>
> We acknowledge the lack of latency and throughput measurements. While EA introduces only two matrix multiplications per layer during scoring and an initial computation of the statistics, we conducted **two additional analyses** following your advice, and added them to Section 4 as we believe they are a valid addition to the method.
>
> 1. We provide a **FLOPs analysis** performed following **[1]**. We compute the model FLOPs for Llama3.1 and the FLOPs resulting from Expected Attention overhead. Results show that this accounts for just **0.5% of the model FLOPs**, a **negligible computational overhead**.
>
> 2. To support this with empirical data, we include **new latency measurements** obtained using our released PyTorch implementation at a 50% compression ratio with Llama3.1-8B. We show results in seconds here below and in the revised Section 4.
>    - **Prefill latency increases by only ~3%**
>    - **Generation latency decreases by ~25%** due to smaller KV Cache.
>
>
> | Phase | No Compression | EA (50% comp.) | Variation (abs) | Variation (%) |
> | :--- | :---: | :---: | :---: | :---: |
> | Prefilling | 15.25 $\pm$ 0.02 | 15.52 $\pm$ 0.02 | +0.27 | +1.74 |
> | Generation | 4.33 $\pm$ 0.00 | 3.22 $\pm$ 0.04 | -1.11 | -25.58 |
> | **Total** | 19.58 $\pm$ 0.03 | 18.74 $\pm$ 0.03 | -0.84 | **-4.30** |
>
> We believe these results are enough to **validate the computational efficiency of Expected Attention**, confirming its negligible theoretical overhead and demonstrating a significant empirical speedup in the generation phase. **The overall inference time is reduced by over 4%.**
>
> Also, we point out that these numbers serve as an upper bound since optimized kernels (e.g., FlashAttention-style fused ops) would reduce overhead further. We added these results in Sec 4.4 in the main paper as we believe they are a valid addition to the method.
>
> **3. Missing and Incorrect Figure Captions**
>
> Thank you for spotting these issues.
> - We added the missing caption in Figure 5.
> - We moved the caption in Figure 4 below the figure as per ICLR rules.

---

> ### Author Response · Authors · 2025-11-21
> **Reply (2/2)**
>
> ### Questions
>
> **All reported experiments appear to use batch size 1, focusing on single-sequence inference. In real-world deployment scenarios, LLM inference often runs with larger batch sizes for better throughput and GPU utilization. How would Expected Attention perform when scaling batch size — both in terms of GPU memory consumption and inference latency — compared to prior training-free baselines? Evaluating this aspect would provide stronger evidence of the method’s practicality for real-world applications.**
>
> EA is fully compatible with batch size > 1 during the scoring phase, as the scoring factorizes across tokens. However, our goal in this work is to study the research question in terms of accuracy, theory, and behavior of a principled scoring mechanism rather than to optimize deployment configuration. For this reason, we focused on the long-context, single-sequence regime, which is both the most impacted by KV growth and the most relevant for evaluating the scientific contribution of EA. However, future engineering-centric work can be dedicated to optimizing EA for high-throughput batched inference, including the implementation of specialized kernels and full evaluation across various batch sizes.
>
> **How does Expected Attention compare to other training-free KV cache compression methods in terms of actual GPU memory consumption, inference latency, and throughput? Including such comparisons would clarify the real-world efficiency benefits of the method.**
>
> Some prior methods **[2]** require specialized kernel support or introduce significant runtime overhead **[3]**. EA avoids this: it adds only two lightweight matrix multiplications for scoring, making its overhead comparable to the simplest scoring-based baselines after computing the statistics.
> As addressed in the response to Weakness 2, we have provided a detailed FLOPs analysis (negligible overhead) and empirical latency measurements (showing a significant generation speedup) which directly validate the efficiency of EA. Regarding memory, we pointed out that under matched compression ratios, the memory footprint for all training-free KV-compression methods is identical. Our value lies in achieving higher accuracy at that matched memory budget.
>
> Finally, we would like to point out that together with the paper, we are releasing **a comprehensive benchmarking library that reproduces all scores for the baselines and our method**. This ensures transparency and lets reviewers inspect the exact implementation. We hope this library can serve as a standard reference for KV Cache compression research in the future.
>
> ---
> Thank you for your comments, we believe the addition that we made to the paper following your concerns improved our contribution significantly. We remain available for further clarification or discussion if needed.
>
>
> ### **References**
> **[1]** Jordan Hoffmann et al., **"Training Compute-Optimal Large Language Models"** in Advances in Neural Information Processing Systems (2022).
>
> **[2]** Yuan Feng et al., **"Ada-KV: Optimizing KV Cache Eviction by Adaptive Budget Allocation for Efficient LLM Inference"**, in Advances in Neural Information Processing Systems 35 (2025).
>
> **[3]** Jang-Hyun Kim et al., **"KVzip: Query-Agnostic KV Cache Compression with Context Reconstruction"** in ES-FoMo III: 3rd Workshop on Efficient Systems for Foundation Models (2025).

---

> ### Comment · Area_Chair_7B1b · 2025-11-23
>
> Dear reviewer,
>
> Thanks for your time and effort in reviewing ICLR2026 submissions. The authors have submitted their responses to your review. Please take the time to read and raise your further comments, and discuss with the authors.
>
> Best regards,
>
> AC

---

### Author Response · Authors · 2025-11-21
**Answer to reviewers**

We thank all reviewers for their thoughtful and constructive feedback. We are encouraged that reviewers found our method **novel** (R5kT, 6DBQ, Vi4u), **theoretically grounded** (R5kT, Vi4u), **simple and intuitive** (myPE, 6DBQ), and **supported by extensive empirical validation** (R5kT, Vi4u). The reviews have highlighted important areas for improvement, particularly regarding computational efficiency analysis, experimental completeness, and clearer presentation of our theoretical assumptions.


Based on your feedback, we have made the following substantial additions to the manuscript (highlighted in orange):

### 1. Computational Efficiency Analysis

We added **efficiency measurements** including:

- **FLOPs analysis** showing EA introduces only **0.5% overhead** relative to model computation
- **Empirical latency measurements** on Llama3.1-8B at 50% compression showing **~3% prefill overhead** and **~25% generation speedup**

### 2. Extended Experimental Results

- **Complete RULER sub-task breakdown** including Multi-key 3 retrieval (Appendix D)
- **Ablation studies on hyperparameter sensitivity** (T window size) in Section 5
- **Ablation studies on covariance**  in Section 5
- **Ablation studies on T window size**  in Section 5
- **Ablation studies on adaptive compression** in Section 5

### 3. Other changes
- We corrected a mistake in the hyper-parameter for the buffer size. We reported 128 but the actual one used for experiments is 256. This does not affect the conclusions of the paper, and we updated the paper accordingly.
- We clarified our modeling assumptions in Section 2.2 and analysed in detail some of the experimental results in Section 4.1 and Appendix D.

### **Code and Reproducibility**
Finally, we point out that we are releasing a **comprehensive benchmarking library** with our submission that **reproduces all reported results**, ensuring full transparency and providing a potential standard reference for future KV cache compression research.

We are grateful to the reviewers for their feedback and welcome any further questions or discussion.

---

### Author Response · Authors · 2025-12-02
**Discussion summary for the new AC**

We had a productive discussion phase, and the improvements made to the paper led one reviewer to raise their scores (4 $\rightarrow$ 6). Given that the discussion, to the best of our knowledge, was not influenced by the OpenReview leak, we are very disappointed to see both the rebuttal responses and the score changes rolled back.

We regret that the data breach prevented the discussion from continuing, as we believe other reviewers would have engaged constructively with our responses, potentially leading to further positive updates to their assessments. Despite this setback, we have continued refining the paper and have updated a third revision that now includes additional ablations on the impact of values norms. We will keep further improving the work following the reviewers' suggestions from the discussion phase.

We hope the new AC considers the original, largely positive assessments of our work, together with the substantial improvements made during the discussion phase.

---

### Meta-Review · Area_Chair_f3hg · 2026-01-08

**Summary:**

This paper presents Expected Attention, a training-free approach for compressing the KV cache during large language model inference. It estimates each key’s expected contribution to future attention by assuming hidden states follow a Gaussian distribution and analytically computing expected attention scores using moment-generating functions. Experiments on long-context and reasoning benchmarks show its effectiveness.

The main concerns are the following:
1. The experiments part are not so convincing, with regard to the comparison with some existing method (TOVA) does not show advantages, and the choice of window size T, the sequence length for RULER is limited. While some of them are discussed and addressed in the rebuttal, overall its hard to evaluate that the expected attention is an useful KV cache compression in practice.
2. The computational efficiency analysis is very limited even considering the additions in the rebuttal. The FLOPs analysis does not imply gain in real implementation, one would like to see an end-to-end efficiency performance. On the other hand, I am quite surprised that the prefilling and decoding for Llama 3.1-8B need 15s and 3-4s respectively, which is much slower than real case (hundreds tokens per second). I'm not saying table 3 is wrong, but such different data can not convincingly support the claims on latency analysis.
3. Most importantly, I'm quite curious that no reviewer mentioned the concern that the paper does not include a comparison of the proposed expected attention score with the original score, either from the mathematical derivation or empirical tests. ONLY with benchmark  dataset experiment(Longbench and so on) is not sufficient.

Overall, I do agree the proposed expected attention idea is novel and interesting, but the limited theoretical analysis and experiments did not show enough evidence for the practical effectiveness of this method.

**Reviewer Concerns:**

Please refer above summary the concerns that were not addressed.

**Reviewer Scores:**

The original scores are 6(Vi4u), 4(6DBQ), 4(myPE), 6(R5kT). I would expect a change to 6,6,4,6.

---

### Decision · Program_Chairs · 2026-01-26

Reject